# Horizontal transfer of the *rfb* cluster in *Leptospira* is a genetic determinant of serovar identity

Cecilia Nieves[1] , Antony T Vincent[1,2], Leticia Zarantonelli[3] , Mathieu Picardeau[4,5], Frédéric J Veyrier[1,*] , Alejandro Buschiazzo[3,5,*]

***Leptospira* bacteria comprise numerous species, several of which cause serious disease to a broad range of hosts including humans. These spirochetes exhibit large intraspecific variation, resulting in complex tabulations of serogroups/serovars that crisscross the species classification. Serovar identity, linked to biological/clinical phenotypes, depends on the structure of surface-exposed LPS. Many LPS biosynthesis–encoding genes reside within the chromosomic *rfb* gene cluster. However, the genetic basis of intraspecies variability is not fully understood, constraining diagnostics/typing methods to cumbersome serologic procedures. We now show that the gene content of the *rfb* cluster strongly correlates with *Leptospira* serovar designation. Whole-genome sequencing of pathogenic *L. noguchii*, including strains of different serogroups, reveals that the *rfb* cluster undergoes extensive horizontal gene transfer. The *rfb* clusters from several *Leptospira* species disclose a univocal correspondence between gene composition and serovar identity. This work paves the way to genetic typing of *Leptospira* serovars, and to pinpointing specific genes within the distinct *rfb* clusters, encoding host-specific virulence traits. Further research shall unveil the molecular mechanism of *rfb* transfer among *Leptospira* strains and species.**

## Introduction

Leptospirosis is a bacterial disease that affects humans and animals. Despite being one of the most extended zoonoses worldwide, leptospirosis remains a neglected and underdiagnosed febrile illness. Pathogenic *Leptospira* species, the etiological agents of leptospirosis, infect a broad spectrum of hosts, with a global annual incidence of 1 million human cases and ~60,000 deaths (Costa et al, 2015; Torgerson et al, 2015). Leptospirosis constitutes an important case model within the "One Health" perspective (Jancloes et al, 2014), being a zoonotic disease that spreads among symptomatic and asymptomatic hosts, with transmission strongly influenced by environmental conditions (Mwachui et al, 2015).

Spirochetes belonging to the genus *Leptospira* have traditionally been divided into three groups: pathogens, intermediates, and saprophytes (Ko et al, 2009). This classification relied on bacterial virulence, isolation from infected hosts, and phylogeny. Recently, expanded and more elaborate phylogenetic analyses resulted in a comprehensive new classification scheme, beyond the species' infectious capacity. The 68 species of *Leptospira* that have been identified so far (Vincent et al, 2019; Korba et al, 2021) are thus classified into two major clades: P (pathogenic) and S (saprophytic), each of which subdivided into two subclades (Vincent et al, 2019).

*Leptospira noguchii* belongs to the first subclade within the P group (P1), which comprises the most important species causing human and animal diseases such as *L. interrogans* and *L. borgpetersenii* (Vincent et al, 2019). Many human infections by *L. noguchii* have been recorded since 1940 (Gochenour et al, 1952; Fraser et al, 1973), but it was not until 1987 that it was recognized as a distinct species (Yasuda et al, 1987), baptized after the Japanese bacteriologist Hideyo Noguchi, himself responsible for choosing the genus name. Despite the clinical importance of *L. noguchii* and its extended geographical distribution, it has received far less attention compared with other pathogenic *Leptospira* species. Noticeably, no finished or closed whole-genome sequence (WGS) of *L. noguchii* is currently available, data that would otherwise boost the power of comparative genomics analyses.

Why could *L. noguchii* WGS contribute with novel insights into leptospirosis? Comprising the largest reported genome among *Leptospira* spp. (Fouts et al, 2016), *L. noguchii* also ranks among the species with more genes and predicted proteins of the entire genus, particularly compared with the closely related pathogens *L. interrogans* and *L. borgpetersenii*, both of which have been studied

[1]Bacterial Symbionts Evolution, Centre Armand-Frappier Santé Biotechnologie, Institut National de la Recherche Scientifique, Université du Québec, Laval, Canada [2]Département des Sciences Animales, Faculté des Sciences de l'agriculture et de l'alimentation, Université Laval, Quebec City, Canada [3]Laboratory of Molecular and Structural Microbiology, Institut Pasteur de Montevideo, Montevideo, Uruguay [4]Institut Pasteur, Université Paris Cité, CNRS UMR 6047, Biology of Spirochetes Unit, Paris, France [5]Integrative Microbiology of Zoonotic Agents, Pasteur International Joint Research Unit, Paris/Montevideo, France/Uruguay

Correspondence: frederic.veyrier@inrs.ca; alebus@pasteur.edu.uy
*Frédéric J Veyrier and Alejandro Buschiazzo contributed equally to this work
Leticia Zarantonelli's present address is Unidad Mixta Pasteur INIA, Institut Pasteur de Montevideo, Montevideo, Uruguay

with much more detail. Besides being isolated from humans, *L. noguchii* has been isolated from armadillos, cattle, sheep, dogs, frogs, and opossums, among others (Silva et al, 2007, 2009), demonstrating its remarkably high adaptability to infecting a very broad range of hosts. Human infections by *L. noguchii* have been reported in geographic areas where the same strains were previously detected in other hosts (Silva et al, 2009; Flores et al, 2017), confirming *L. noguchii*'s capacity for zoonotic transmission. Found predominantly in the Americas, and more rarely in Asia (Guglielmini et al, 2019), *L. noguchii* exhibits high genetic diversity among circulating strains (Martins et al, 2015; Hamond et al, 2016; Zarantonelli et al, 2018), with no apparent correlation between genotypes and hosts or geographic distribution (Loureiro et al, 2020). Particularly in South America, systematic field studies of infected animal hosts reveal a much larger diversity of *L. noguchii* serovars than that encountered for *L. interrogans* and *L. borgpetersenii* (Zarantonelli et al, 2018).

The presence of many serovars (serologic variants) is a common attribute within *Leptospira* species. With >300 serovars having been reported, their classification into serogroups has been instrumental, clustering together related serovars that express overlapping antigenic determinants (Bharti et al, 2003). Probably related to variable structures of the surface-exposed LPS antigen on the bacterial cell wall (Adler & de la Peña Moctezuma, 2010), different serovars trigger distinct antibody responses during infection. Interestingly, a number of known serovar–host associations have been pinpointed, leading to the concept of serovar adaptation and chronicity of infection for particular hosts, also correlating to more acute virulence when non-adapted serovars accidentally infect heterologous hosts (Ellis, 2015). Despite the relevance of this phenomenon in terms of epidemiology and clinical outcomes, the molecular mechanisms that underlie serovar determination are not fully understood (de la Peña-Moctezuma et al, 1999; Adler & de la Peña Moctezuma, 2010; Fouts et al, 2016). A connection between serovar determination and gene content has been proposed (Bulach et al, 2000, 2006; Santos et al, 2018), but not demonstrating a direct, biunivocal link among each of the many different serovars and a defined set of genes (genetic presence/absence profiles). Such unequivocal link has also been hampered by the scarcity of precise serovar identification for most reported isolates, and the lack of finished whole-genome sequencing data, particularly so for *L. noguchii* and other understudied species.

By sequencing the genomes of 12 *L. noguchii* strains (10 closed genomes and 2 drafts), we undertook an extensive comparative genomics approach, uncovering underlying reasons for *Leptospira* phenotypic complexity. We now reveal (i) the detailed genomic features and plasmid repertoire of *L. noguchii* and its phylogenetic structure; (ii) that the cluster comprising most of the LPS synthesis enzyme-encoding genes, known as *rfb* (Patra et al, 2015; Picardeau, 2017), exhibits clear signs of horizontal gene transfer (HGT) among different *Leptospira* species; and (iii) that serovar identity is univocally linked to the presence/absence of specific genes within this *rfb* cluster.

In sum, this work constitutes the first report of complete genomes of *L. noguchii*, which allowed a comprehensive analysis of its genetic variability. Remarkably, after comparing with known serovars of different *Leptospira* species, it was possible to reveal

serovar-specific genetic fingerprints encoded within a horizontally transferred gene cluster, paving the way toward genome-based serotyping and further molecular studies of the HGT mechanisms at play.

# Results

## *L. noguchii* WGSs: general features

Whole genomes from 12 *L. noguchii* strains (Table 1) were sequenced using a long-read sequencing approach (PacBio technology). These strains were isolated from different hosts at four distant geographic locations in Central and South America: Barbados (two isolates from amphibian hosts), Guadeloupe island/France (one, human), Uruguay (eight, cattle), and Venezuela (one, human). Exhibiting an average genome size of 4,863,036 ± 99,185 bp, all were larger than other well-studied species such as *L. interrogans* (~4.6 Mb) and *L. borgpetersenii* (~3.9 Mb). Most genomes reached a finished status, with three to eight contigs (Table 1) corresponding to chromosomes 1 (Chr1) and 2 (Chr2), plus a variable number of plasmids. The whole genomes from strains "bajan" and "201102933" were the only ones not closed, albeit rendering very high-quality draft sequences (five contigs in the case of strain bajan, and six for strain 201102933, with N50s of 2,827,750 and 3,551,682 bp, respectively).

The average nucleotide identity (ANI) among all sequenced strains in this study (using *L. noguchii* sv Panama strain CZ214 as reference) was >95% (Table S1 and Fig S1; see all Supplemental Material in Supplemental Data 1), consistent with previous determinations based on 16S rDNA sequence (Zarantonelli et al, 2018). Similar identity figures with all reported draft WGSs from *L. noguchii* strains further confirm the taxonomic determination, and ANIs ≤90% with respect to closely related species such as *L. interrogans* and *L. kirschneri*. However, the percentage of conserved proteins among the 12 sequenced strains did not exceed 98.1% (Fig S2), uncovering a significant intraspecies phenotypic complexity. Especially diverse in terms of protein repertoire is the cluster comprising the three Caribbean strains (bajan, barbudensis, and 2011029331), which consistently exhibit highest ANI figures among them. Regarding replicon content, *L. noguchii* displays the same genetic organization as reported for other *Leptospira* species (Picardeau et al, 2008), with two chromosomes and a variable number of plasmids (Table 1). Consistent with their larger genome size, *L. noguchii* also exhibits more CDSs (~4,000 on average) compared with other *Leptospira* species (Picardeau et al, 2008), with other features such as number of tRNA genes and GC content being similar. The number of predicted CRISPR sequences was more variable, as were those of transposases and IS transposase-like CDSs, ranging from 95 to 145 (Table 1), in any case much more numerous than in *L. interrogans* (26 in sv Copenhageni strain Fiocruz) or in the saprophyte *Leptospira biflexa* (8 in sv Patoc strain Ames) (Picardeau et al, 2008).

Analysis of the *L. noguchii* pangenome of the 12 strains sequenced in this study (Fig S3) showed an open profile (Heaps' law parameter $\alpha$ = 0.36 [Tettelin et al, 2008]), confirming the high genetic variability among *L. noguchii* strains. Out of the 7,963 genes that constitute the pangenome, only 2,671 were found in almost all the

**Table 1.** *L. noguchii* whole-genome sequences.

| | Strain | Host | Replicon | Size (bp) | CDSs | rRNA | tRNA | CRISPRs | TPases | GC (%) | Coding ratio (%) | Accession number |
|---|---|---|---|---|---|---|---|---|---|---|---|---|
| Finished genomes | | | | | | | | | | | | |
| | barbudensis (AUS/BRB) | Amphibian | Chr1 | 4,408,823 | 3,883 | 5 | 37 | 6 | 101 | 35.5 | 75.9 | CP091967 |
| | | | Chr2 | 359,178 | 337 | — | — | — | 3 | 35.3 | 76.8 | CP091968 |
| | | | p1 | 47,641 | 65 | — | — | — | 0 | 42.2 | 45.2 | CP091969 |
| | | | p2 | 44,440 | 63 | — | — | — | 0 | 42.5 | 46 | CP091970 |
| | | | Total | 4,860,082 | | | | | | | | |
| | IP1512017 (ND/URY) | Cattle | Chr1 | 4,297,194 | 3,375 | 5 | 37 | 8 | 104 | 35.6 | 73.3 | CP091957 |
| | | | Chr2 | 370,404 | 327 | — | — | — | 14 | 36 | 77.1 | CP091958 |
| | | | p1 | 178,101 | 139 | — | — | — | 7 | 38.3 | 74.3 | CP091959 |
| | | | p2 | 67,330 | 48 | — | — | — | 3 | 37.2 | 76.6 | CP091960 |
| | | | p3 | 43,769 | 33 | — | — | — | 0 | 35.4 | 73 | CP091961 |
| | | | Total | 4,956,798 | | | | | | | | |
| | IP1605021 (PYR/URY) | Cattle | Chr1 | 4,438,826 | 3,555 | 5 | 37 | 10 | 121 | 35.8 | 73.4 | CP091953 |
| | | | Chr2 | 395,164 | 368 | — | — | — | 10 | 36.2 | 79.5 | CP091954 |
| | | | p1 | 169,819 | 136 | — | — | — | 11 | 38.1 | 75 | CP091955 |
| | | | p2 | 47,268 | 29 | — | — | — | 3 | 36.4 | 67.3 | CP091956 |
| | | | Total | 5,051,077 | | | | | | | | |
| | IP1611024 (AUS/URY) | Cattle | Chr1 | 4,248,034 | 3,329 | 5 | 37 | 10 | 83 | 35.6 | 74 | CP091947 |
| | | | Chr2 | 351,624 | 291 | — | — | — | 10 | 35.6 | 76.6 | CP091948 |
| | | | p1 | 89,758 | 93 | — | — | — | 0 | 32.4 | 83.1 | CP091949 |
| | | | p2 | 74,081 | 60 | — | — | — | 5 | 34.6 | 67 | CP091950 |
| | | | p3 | 57,447 | 50 | — | — | — | 3 | 34.3 | 61.2 | CP091951 |
| | | | p4 | 41,294 | 30 | — | — | — | 1 | 32.4 | 71.2 | CP091952 |
| | | | Total | 4,862,238 | | | | | | | | |
| | IP1703027 (ND/URY) | Cattle | Chr1 | 4,329,282 | 3,408 | 5 | 37 | 12 | 94 | 35.7 | 73.7 | CP091943 |
| | | | Chr2 | 338,457 | 275 | 0 | 0 | 1 | 3 | 35.7 | 76.4 | CP091944 |
| | | | p1 | 83,272 | 52 | — | — | — | 1 | 35.9 | 71.2 | CP091945 |
| | | | p2 | 61,325 | 44 | — | — | — | 3 | 35.7 | 73.1 | CP091946 |
| | | | Total | 4,812,336 | | | | | | | | |
| | IP1705032 (AUT/URY) | Cattle | Chr1 | 4,342,451 | 3,435 | 5 | 37 | 10 | 98 | 35.7 | 74 | CP091940 |
| | | | Chr2 | 343,634 | 286 | — | — | — | 4 | 35.6 | 77.3 | CP091941 |
| | | | p1 | 58,982 | 49 | — | — | — | 0 | 35.9 | 66.1 | CP091942 |
| | | | Total | 4,745,067 | | | | | | | | |
| | IP1709037 (AUT/URY) | Cattle | Chr1 | 4,199,394 | 3,354 | 5 | 37 | 8 | 81 | 35.7 | 74.1 | CP091936 |
| | | | Chr2 | 376,569 | 326 | — | 1 | — | 3 | 36.1 | 76.8 | CP091937 |
| | | | p1 | 90,518 | 74 | — | — | — | 6 | 35.3 | 73.5 | CP091938 |
| | | | p2 | 74,070 | 61 | — | — | — | 5 | 34.5 | 65.7 | CP091939 |
| | | | Total | 4,740,551 | | | | | | | | |

**Table 1. Continued**

| | Strain | Host | Replicon | Size (bp) | CDSs | rRNA | tRNA | CRISPRs | TPases | GC (%) | Coding ratio (%) | Accession number |
|---|---|---|---|---|---|---|---|---|---|---|---|---|
| | IP1712055 (ND/URY) | Cattle | Chr1 | 4,329,115 | 3,455 | 5 | 37 | 12 | 93 | 35.7 | 73.6 | CP091928 |
| | | | Chr2 | 338,447 | 277 | — | — | 1 | 3 | 35.7 | 76.2 | CP091929 |
| | | | p1 | 83,272 | 52 | — | — | — | 1 | 35.9 | 71.2 | CP091930 |
| | | | p2 | 61,322 | 44 | — | — | — | 3 | 35.7 | 71.5 | CP091931 |
| | | | p3 | 35,952 | 34 | — | — | — | 0 | 41.3 | 53.1 | CP091932 |
| | | | p4 | 26,771 | 7 | — | — | — | 0 | 41.1 | 5.7 | CP091933 |
| | | | p5 | 24,253 | 38 | — | — | — | 0 | 42 | 71.5 | CP091934 |
| | | | p6 | 13,148 | 11 | — | — | — | 0 | 40.3 | 36.8 | CP091935 |
| | | | Total | 4,912,280 | | | | | | | | |
| | IP1804061 (ND/URY) | Cattle | Chr1 | 4,322,667 | 3,447 | 5 | 37 | 9 | 93 | 35.7 | 73.4 | CP092112 |
| | | | Chr2 | 330,637 | 280 | — | — | — | 16 | 35.8 | 76 | CP092113 |
| | | | p1 | 127,018 | 100 | — | — | — | 5 | 37.3 | 70.8 | CP092114 |
| | | | p2 | 100,275 | 71 | — | — | — | 0 | 36.3 | 74.1 | CP092115 |
| | | | p3 | 51,374 | 37 | — | — | — | 2 | 35.2 | 72.1 | CP092116 |
| | | | Total | 4,931,971 | | | | | | | | |
| | 201601331 (ND/VEN) | Human | Chr1 | 4,334,850 | 3,447 | 5 | 37 | 8 | 134 | 35.7 | 74 | CP091962 |
| | | | Chr2 | 350,933 | 293 | — | — | — | 5 | 35.7 | 76.8 | CP091963 |
| | | | p1 | 65,245 | 46 | — | — | — | 1 | 35.9 | 65.2 | CP091964 |
| | | | p2 | 38,153 | 33 | — | — | — | 2 | 33.4 | 57 | CP091965 |
| | | | p3 | 28,484 | 26 | — | — | — | 1 | 35.8 | 34.1 | CP091966 |
| | | | Total | 4,817,665 | | | | | | | | |
| Draft genomes | | | | | | | | | | | | |
| | bajan (AUS/ BRB) | Amphibian | Five (5) contigs | 4,850,434 | 4,326 | 5 | 37 | 6 | 104 | 35.7 | 76.4 | JAKNBP000000000 |
| | 201102933 (AUS/GLP) | Human | Six (6) contigs | 4,697,964 | 3,869 | 5 | 38 | 6 | 103 | 35.5 | 76.9 | JAKNBO000000000 |

The "Strain" column provides information about the serogroup of each strain (AUS, Australis; AUT, Autumnalis; PYR, Pyrogenes; ND, not determined) and its country of origin (BRB, Barbados; GLP, Guadeloupe; URY, Uruguay; VEN, Venezuela), in parentheses. The other columns provide information as described in the column headings. "TPases" stands for transposases.

strains thus comprising the core genome. Indeed, the cloud genome was defined by 2,183 genes (accessory genes), uncovering a rich array of unique attributes that distinguish strains.

To further explore the properties of such strain-specific genes, likely underlying phenotypic variability, profile hidden Markov models were calculated for each one of the genes present in only one of the strains and absent in all others. These hidden Markov model profiles were then mapped (Aramaki et al, 2020) onto the Kyoto Encyclopedia of Genes and Genomes (KEGG) database to investigate whether these variant-specific genes are enriched in particular biochemical functions, or instead randomly distributed (Table S2). Among the four top-ranking pathways—(i) metabolic pathways, (ii) biosynthesis of nucleotide sugars, (iii) amino sugar and nucleotide sugar metabolism, and (iv) O-antigen nucleotide sugar biosynthesis—a clear enrichment is observed in functions related to carbohydrate metabolism, and glycoside modification

and synthesis. This KEGG mapping analysis was systematically extended to accessory genes present only in two strains, three, and further, confirming the importance of variations in carbohydrate-related metabolism as a main source of strain-specific genetic variability. Of note, a number of carbohydrate-related genes, including several that encode LPS biosynthesis enzymes, were present only in particular groups of strains. For example, UDP-glucuronate 4-epimerase (GalE [Bulach et al, 2000]) or 3-deoxy-D-manno-octulosonate 8-phosphate phosphatase (KdsC [Biswas et al, 2009; Valvano, 2015]), among others, clusters according to serogroup identity.

### Plasmid repertoire in *L. noguchii*

Besides the two chromosomes, *L. noguchii* strains harbor a variable number of plasmids (Tables 1 and S3, first sheet), ranging from only

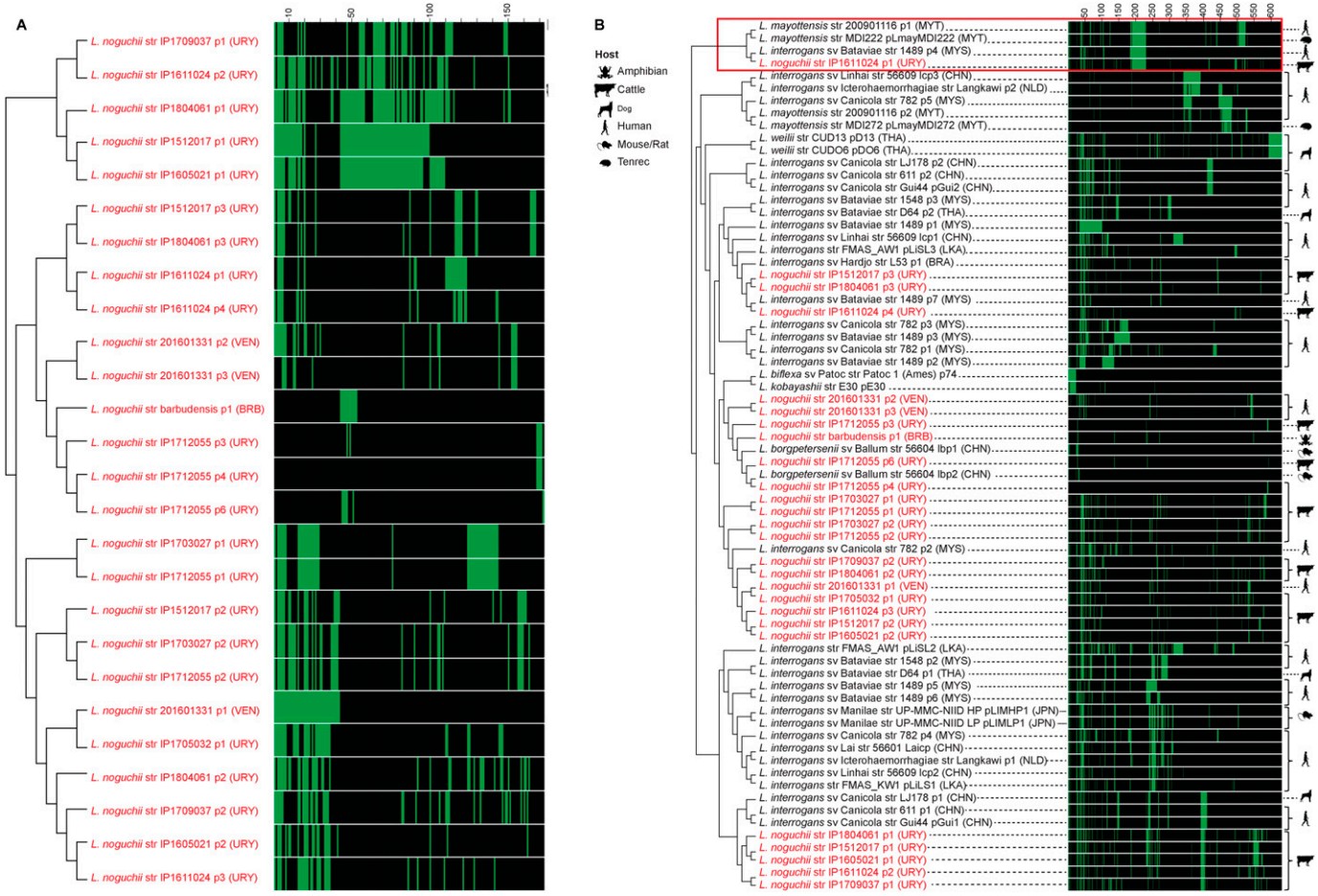

**Figure 1. Conservation of protein-encoding genes across *Leptospira* plasmids.**
Sequence network association analysis by hierarchical clustering, based on the presence/absence of plasmidic protein–encoding genes. Matrices on the right of each clustering depict individual genes with vertical lines, green meaning that the gene is present in that strain (60% similarity cutoff), and black meaning absence. Scales on the top of the matrices indicate the number of different genes being compared. **(A)** Plasmids from *L. noguchii* strains sequenced in this study. Plasmid names are indicated right after the strain designation, and country of origin in parentheses (BRB, Barbados; URY, Uruguay; VEN, Venezuela). **(B)** Plasmids from different *Leptospira* species. Plasmid names as in (A), and country of origin in parentheses (BRA, Brazil; BRB, Barbados; CHN, China; JPN, Japan; LKA, Sri Lanka; MYS, Malaysia; MYT, Mayotte; NLD, the Netherlands; THA, Thailand; URY, Uruguay; VEN, Venezuela). Strains sequenced in this study are highlighted in red. The hosts from which they were isolated are indicated with cartoons on the right side of the matrix. The red square encloses plasmids from different *Leptospira* species sharing a large number of protein-encoding genes.

one to as much as six replicons. The plasmid repertoire is unique to each strain. A network association analysis was performed to compare the plasmid-encoded proteins in different *L. noguchii* strains (Fig 1A). Some plasmids showed identical or nearly identical presence/absence patterns of protein-encoding genes, suggesting that plasmids may be transferred among strains. For example, the two plasmids of strain IP1703027 bear identical gene composition compared with two of the plasmids in IP1712055 (p1 and p2); plasmids p1 from strains IP1512017 and IP1605021, and plasmids p3 from IP1512017 and IP1804061 share many genes.

Extending the analysis of plasmid repertoires to other strains and *Leptospira* species (Table S3, second sheet) revealed no core or even softcore genes, highlighting the extreme plasmid diversity in *Leptospira*. A network association analysis considering this extended set of plasmids revealed species-specific clustering of plasmid sequences, again uncovering cases of strong similarity/identity in the arrays of protein-encoding genes between different

strains (Fig 1B). For instance, plasmids p1/p2, along with p2/p3 from distinct *L. interrogans* strains D64 and 1548, exhibit nearly identical profiles. The same was observed for plasmids p1 and p2 from *L. interrogans* strains 611, Gui44, and LJ178; and for comparing plasmids pD13 and pDO6 from *L. weilii*. Overall, plasmids from *L. noguchii* clustered together and did not show similar patterns to those from other species, except just in one case. Plasmid p1 from strain IP1611024 shared a significant number of protein-encoding genes with plasmids p1 from *L. mayottensis* str 200901116, pLmayMDI222 from *L. mayottensis* str MDI222, and p4 from *L. interrogans* str 1489 (Fig 1B, red square).

A functional/biological analysis of *L. noguchii* plasmid–encoded genes is difficult, as most of the constituent proteins are hypothetical (Table S3, third sheet). A first general inquiry about the potential link between plasmid identity and environmental factors did not result in a clear association. Neither the geographical locations of strains, nor the infected host from which they were isolated, showed clear-cut connections with plasmids and their

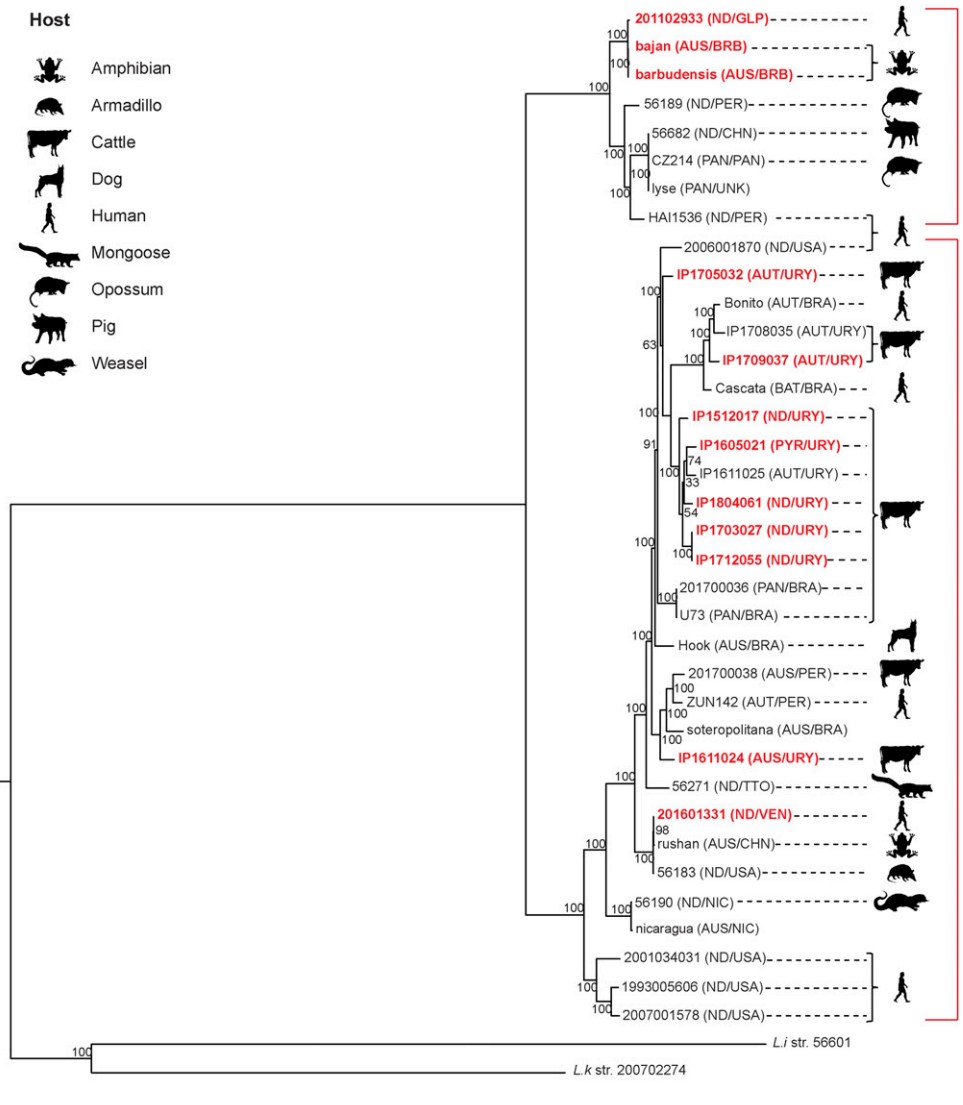

**Figure 2. Phylogenetic tree of *Leptospira noguchii*.**
The maximum-likelihood phylogenetic tree is based on the softcore genes (present in more than 95% of the genomes). *L. interrogans* str 56601 and *L. kirschneri* str 200702274 were included as outgroups. The serogroup of each strain (AUS, Australis; AUT, Autumnalis; BAT, Bataviae; ND, not determined; PAN, Panama; PYR, Pyrogenes) and its country of origin (BRB, Barbados; BRA, Brazil; CHN, China; GLP, Guadeloupe; NIC, Nicaragua; PAN, Panama; PER, Peru; TTO, Trinidad & Tobago; USA, United States of America; URY, Uruguay; UNK, unknown; VEN, Venezuela) are indicated in parentheses. Strains sequenced in this study are outlined in bold red. The hosts from which they were isolated are indicated with cartoons (some hosts were not specified in the original reports). Two clades can be distinguished, highlighted with red brackets.

protein-encoding gene compositions. Plasmid-borne virulence factors and antibiotic resistance determinants were also explored, recognizing two genes encoding putative multidrug efflux proteins of the resistance–nodulation–division family (Nishino et al, 2007), MtdA and MtdB, in plasmids p1 (from strains IP1512017, IP1605021, IP1709037, and IP1804061), and p2 (from IP1611024). The limited identity with bona fide antibiotic resistance proteins did not allow for conclusive antibiotic specificity and/or functional prediction. These genes were always found as a cluster in *Leptospira* plasmids, with *mtdA* followed by *mtdB* and *cusA*, the latter encoding a cation efflux pump. The extended network association analysis including also other *Leptospira* species (Table S3, fourth sheet) revealed this same cluster in plasmid p1 from three *L. interrogans* strains (611, Gui44, and LJ178). Further work is needed to uncover the biological role of these proteins in *Leptospira*, especially considering that antibiotic resistance is not a usual feature in spirochetes. Functional analysis of clusters of orthologous genes (COG) showed a few

categories to be absent from plasmidic genes in *L. noguchii* and other *Leptospira* species: (i) RNA processing and modification; (ii) chromatin structure and dynamics; (iii) carbohydrate metabolism and transport; (iv) nuclear structure; (v) cytoskeleton; and (vi) general function prediction. On the contrary, among the most represented COG categories were those related to (i) unknown function, (ii) replication and repair, (iii) transcription, and (iv) signal transduction. Surprisingly, the functional category linked to amino acid metabolism and transport was completely absent in *L. noguchii* plasmids, in stark contrast to other *Leptospira* species.

### *L. noguchii* phylogeny

The complexity and high diversity of this species' pangenome could also be related to the adaptation of *L. noguchii* to different hosts and geographic locations. To uncover such potential genotype/phenotype associations, the phylogenetic structure of *L. noguchii*

was explored in greater detail by analyzing the genomes from 11 different geographic locations and nine types of hosts (Fig 2 and Table S4) applying a maximum-likelihood approach.

Two clades could be distinguished, which do not correlate to geographic distribution nor to host. Further studies to increase the number of strains shall enable a more conclusive statement. The Uruguayan strains isolated from cattle cluster within one of the clades, but they are not phylogenetically distant from other host species including humans, other mammals, and even amphibians. Focusing the analysis on *L. noguchii* strains isolated from human infections, the distribution is once again extremely broad throughout the phylogenetic tree, indicative of transmission among different reservoirs (Fig 2).

### Genetic variability of the *rfb* cluster in *L. noguchii*

Considering that serologic variability is a particularly relevant phenotypic trait in *Leptospira* (Adler & de la Peña Moctezuma, 2010), that the *L. noguchii* phylogenetic structure did not reveal clear genotype/phenotype associations including host tropism, and that *L. noguchii* strain–specific accessory genes were found to be highly enriched in carbohydrate pathways and LPS biosynthesis, a more detailed analysis focusing on particular genomic regions was done. Genes coding for the cell wall LPS biosynthesis have been linked to serovariation, and in *Leptospira* tend to concentrate within a gene cluster known as *rfb* (de la Peña-Moctezuma et al, 1999; Fouts et al, 2016). Access to complete/finished WGSs is particularly relevant to analyze delimited loci, avoiding inaccurate gene composition descriptions that result from fuzzy boundaries and/or incompleteness (Denton et al, 2014). Exploiting the WGS of *L. noguchii* strains that we are now reporting, a reliable evaluation of gene diversity related to LPS biosynthesis is feasible.

The core moiety of LPS known as lipid A (Raetz & Whitfield, 2002; Que-Gewirth et al, 2004) is synthesized by several enzymes encoded in a cluster of 13 genes: *lpxA, lpxC, lpxD1, lpxD2, lpxB1, lpxB2, lpxK, kdtA, kdsB1, kdsB2, lnt, kdsA*, and *htrB*. The composition of this gene cluster was almost identical in all the strains analyzed, including genomes reported in this work and those from other *Leptospira* strains of known serovar identity (Table S5, first sheet). This result confirms previous reports analyzing several different pathogenic *Leptospira* species (Fouts et al, 2016).

A second component of LPS is the core oligosaccharide (Raetz & Whitfield, 2002), whose biosynthesis starts with the addition of 3-deoxy-D-*manno*-oct-2-ulosonic acid (Kdo) molecules to the lipid A moiety, subsequently incorporating heptoses and further modifications (Bertani & Ruiz, 2018). Comparison of the genes coding for core synthesis enzymes (such as WaaA that adds Kdo molecules; RfaC, RfaD, RfaE, and RfaF that attach ADP-L-*glycero-β*-D-manno-heptose intermediates; and other glycosyltransferases such as RfaG, which append glucose units to the heptoses) among the different strains (Table S5, first sheet) revealed no variation in terms of differential presence of genes. The only difference concerned *L. interrogans* sv Weerasinghe, in which three genes coding for WaaA isoforms were found (~99% identical among them).

Finally, the outermost section of the LPS known as the *O*-antigen is the most variable part of the structure and most exposed to interact with the environment (Raetz & Whitfield, 2002). The

**Table 2. Overall composition of *rfb* clusters in *L. noguchii*.**

| Strain | Size (bp) | CDSs | GC content (%) |
|---|---|---|---|
| IP1512017 (ND/URY) | 84,097 | 74 | 33.7 |
| IP1605021 (PYR/URY) | 113,593 | 110 | 34.0 |
| IP1611024 (AUS/URY) | 106,137 | 97 | 32.8 |
| IP1703027 (ND/URY) | 82,932 | 74 | 33.7 |
| IP1705032 (AUT/URY) | 93,324 | 83 | 31.9 |
| IP1709037 (AUT/URY) | 96,628 | 90 | 32.4 |
| IP1712055 (ND/URY) | 82,927 | 77 | 33.7 |
| IP1804061 (ND/URY) | 82,821 | 74 | 33.7 |
| 201102933 (ND/GLP) | 103,442 | 100 | 33.8 |
| 201601331 (ND/VEN) | 99,122 | 89 | 32.2 |
| bajan (AUS/BRB) | 104,988 | 105 | 33.8 |
| barbudensis (AUS/BRB) | 103,476 | 103 | 33.8 |

The "Strain" column informs, in parentheses, about the serogroup of each strain (AUS, Australis; AUT, Autumnalis; PYR, Pyrogenes; ND, not determined) and its country of origin (BRB, Barbados; GLP, Guadeloupe; URY, Uruguay; VEN, Venezuela).

*O*-antigen is an oligosaccharide comprising a fairly large number of diverse monosaccharides, synthesized and oligomerized together by the action of several enzymes, most of which are encoded in the *rfb* cluster in *Leptospira* (Mitchison et al, 1997). The genetic composition of the *rfb* clusters of the 12 *L. noguchii* genomes revealed a striking variability (Table 2). High *rfb* variability had previously been described comparing different *Leptospira* species (Fouts et al, 2016), hereby confirmed within a single species.

Further insights were obtained by aligning the *rfb* clusters of these 12 *L. noguchii* strains, highlighting the synteny among their constituent genes (Fig 3A).

Delimiting the boundary ends of the *rfb* cluster, genes coding for a transcriptional regulator (MarR) and a sodium/sulfate symporter (SdcS) were consistently found, as in other pathogenic *Leptospira* species (Fouts et al, 2016). The 3′ end is more conserved; toward this end, a gene subcluster is located, composed of *rfbC, rfbD, rfbB*, and *rfbA*, which encodes enzymes involved in the dTDP-rhamnose biosynthesis, implicated in LPS assembly in pathogenic strains (Mitchison et al, 1997). The order of appearance of these four genes was conserved in all strains. Only strain IP1605021 (serogroup Pyrogenes) presented an extra copy of *rfbC* within the *rfb* cluster but separated from the dTDP-rhamnose biosynthesis subcluster. Systematically, the *rfb* cluster was found in preferential locations within Chr1, approximately at ~1.75 and ~2.50 Mb from the origin (Fig 3B), and run in opposite senses comparing locations 1 versus 2. The regions that flank the *rfb* clusters are conserved (Fig 3B), although this depends on the location site. An interesting example of this is illustrated in cases where the same *rfb* cluster is identified in one genomic site or the other in different strains (e.g., strains IP1705032 versus IP1709037, or IP1804061 versus IP1703027; see Fig 3B). A detailed examination of both *rfb* locations revealed an explanation to this feature: genomic inversions are coincident with the *rfb* clusters being located at one site or the other (Fig 3C). Although

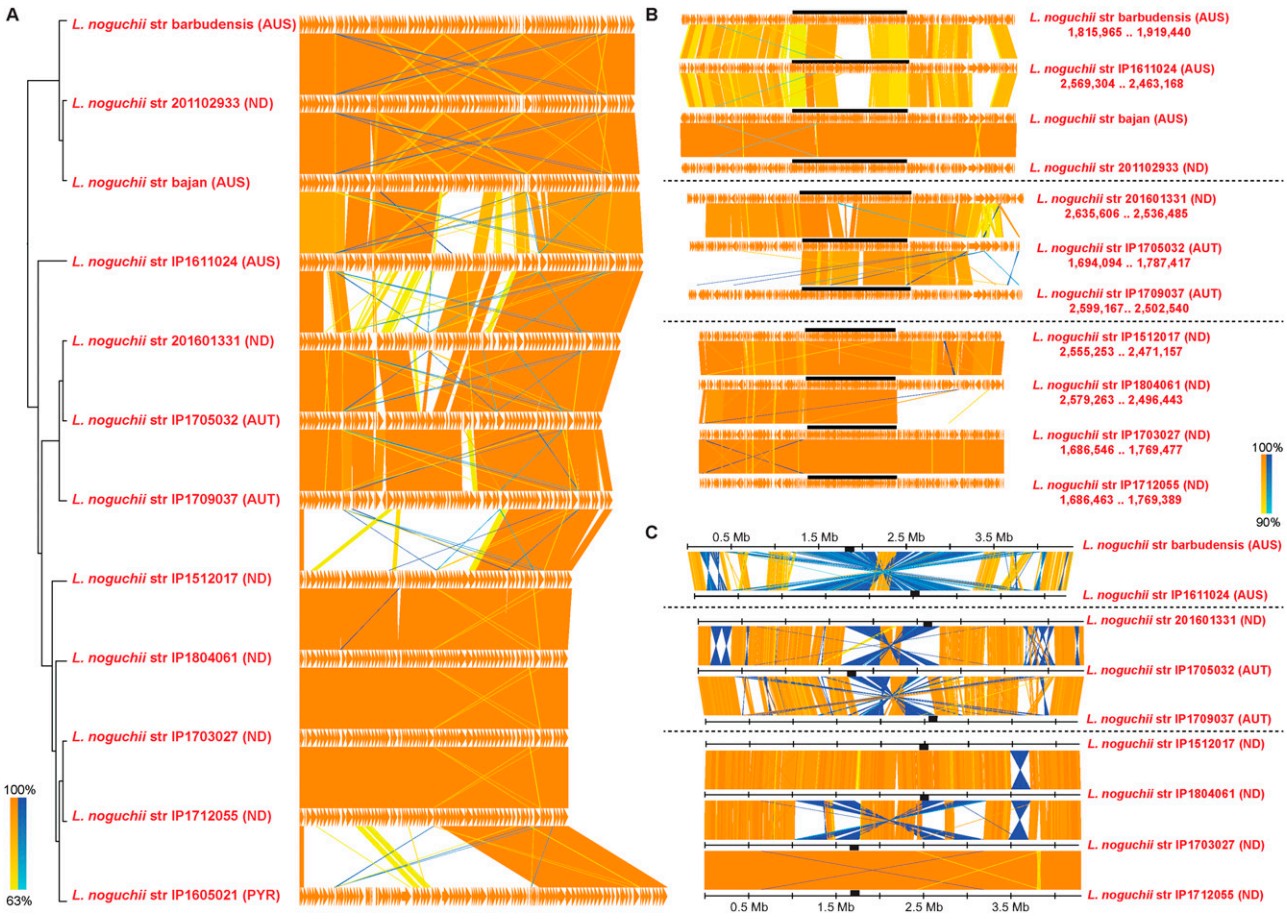

**Figure 3. Comparison of *rfb* clusters in *L. noguchii* shows highly variable gene composition.**
**(A)** softcore *rfb* genes from *L. noguchii* strains aligned and clustered considering a 60% identity cutoff and gene presence in at least 60% of the *rfb* clusters analyzed (left). Serogroup identity is indicated in parentheses (AUS, Australis; AUT, Autumnalis; PYR, Pyrogenes; ND, not determined). To the right, *rfb* cluster gene content and shared synteny are depicted. Homologous regions are linked with orange lines (same orientation) and blue lines (inverted regions), from lighter to darker colors according to identity level (as marked by the lower left scale index). **(B)** Close-up of *rfb* clusters (highlighted as black bars) from *L. noguchii* strains that belong to the same serogroup, or with highly similar *rfb* clusters, including 100,000 extra bp flanking at each side. The genomic coordinates of *rfb*-delimiting genes *marR* and *sdcS* (in base pairs, with *dnaA* at position 0) are indicated below the strains' names for those strains with closed/finished genomes, and their delimitations within the alignments are highlighted as black bars. Nucleotide alignment was performed considering a 90% identity cutoff. Homologous regions are linked with orange lines (same orientation) and blue lines (inverted regions), from lighter to darker colors according to identity level (as marked by the lower right scale index). **(C)** Nucleotide alignments (90% identity cutoff) of entire chromosome 1 from *L. noguchii* strains that possess highly similar *rfb* clusters (barbudensis versus IP1611024; 201601331 versus IP1705032/ IP1709037; and IP1512017 versus IP1804061/IP1703027/IP1712055). Color references as in (B). The *rfb* clusters are highlighted as black bars.

larger in some genomes, such inversions do not implicate the entire genomic range between ~1.75 and ~2.50 Mb, wherein colinear regions are also observed. Interestingly, in some of the strains the *rfb* cluster is located precisely at the boundary where the inversion occurs, thereby explaining why on those cases there is only one conserved *rfb* flanking region (Fig 3B). Of note, insertion sequence (IS) transposase-like CDSs were found at or near the boundaries of the *rfb* cluster (Fig S4 and Table S5, fourth and fifth sheets). Genomic rearrangements involving inversions have been reported in other *Leptospira* species (Nascimento et al, 2004; Olo Ndela et al, 2021), some of which are indeed IS-mediated (Nascimento et al, 2004).

### The *rfb* cluster shows hallmarks of HGT

Signs of HGT were readily identified when analyzing the *rfb* cluster of genes in *L. noguchii*, and in other *Leptospira* species. A significant

decrease in the GC content was systematically observed at the cluster position in all *L. noguchii* genomes reported in this study (Fig 4A). Moreover, the deviated GC content that identifies *rfb* clusters as islands within *L. noguchii* Chr1 was further confirmed by extending these analyses to 10 additional *Leptospira* species for which complete WGSs are available (Fig S5). A conspicuously low-GC-content region corresponds with the position of the *rfb* cluster in all cases, in several of the species being the only such deviated segment, whereas additional ones are present in other cases as well. *L. interrogans* is the species that displays the least pronounced decrease, although the deviation is still evident. In *L. biflexa*, a second nearby segment exhibits a noticeable GC content decrease other than the *rfb* cluster itself (Fig S5).

A second feature pointing to HGT of *rfb* is the decreased sequence conservation of flanking regions. This is more difficult to observe when only *L. noguchii* genomes are compared (Fig 3B and

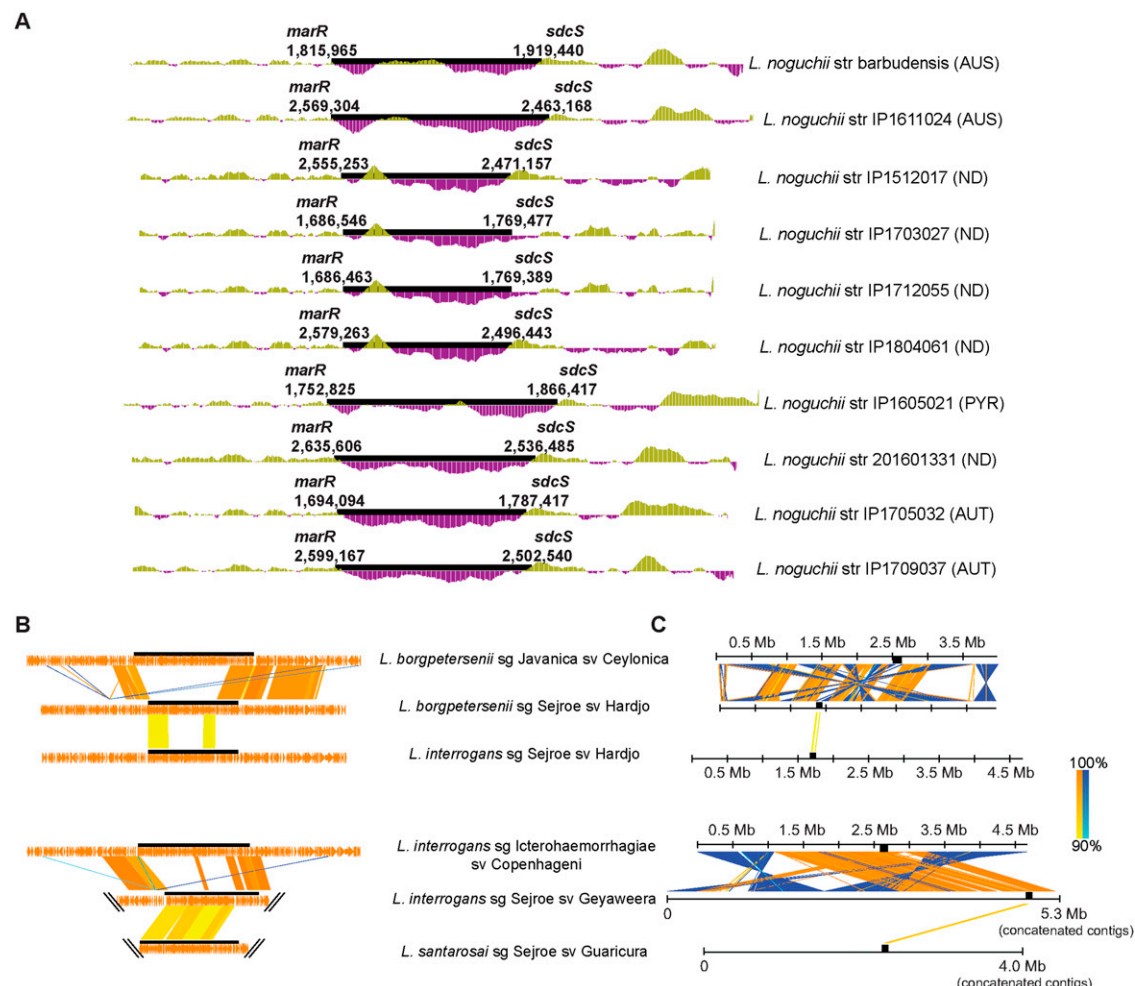

**Figure 4. Genomic details of the *rfb* cluster in *Leptospira* reveal hallmarks of HGT.**
**(A)** Close-up of each of the *rfb* clusters (shown as black bars), including 100,000 extra bp flanking at each side. GC % (calculated every 1,000 bp) is plotted, with purple or green indicating, respectively, a reduced or increased percentage compared with the average found in the whole chromosome 1. Relative position of genes *marR* and *sdcS* delimiting the *rfb* cluster is indicated in base pairs (*dnaA* at the origin). **(B)** Nucleotide alignment (90% identity cutoff) of the *rfb* clusters (black bars) and 100,000 extra base pairs flanking at each side, comparing more distant species all belonging to different serovars within serogroup Sejroe (*L. borgpetersenii* sv Hardjo str L550, *L. interrogans* sv Hardjo str Hardjoprajitno, *L. interrogans* sv Geyaweera str 1L-int, and *L. santarosai* sv Guaricura str M4/98). Unrelated serovars (*L. borgpetersenii* sv Ceylonica str Piyasena and *L. interrogans* sv Copenhageni str Fiocruz L1-130) were also added to compare species-specific conservation, outside of the *rfb* cluster. Homologous genes are linked with orange lines (if they share the same orientation) and blue lines (for inverted orientations), from lighter to darker colors according to identity percentage as marked by the right scale index. Of note, the genomes from *L. interrogans* sv Geyaweera and *L. santarosai* sv Guaricura are draft genomes. In these cases, contig boundaries are indicated with parallel black slashes, as the analysis could not be reliably extended beyond those limits. **(C)** Nucleotide alignment (90% identity cutoff) over the entire chromosome 1 (depicted as straight black lines), comparing the same set of species as in (B). The position of *rfb* clusters is indicated with black square blocks. Reference colors as in (B).

C), because of the overall higher conservation within a species. However, the *rfb* flanking regions' variability becomes evident when comparing different species of the same serovar and serogroup. In these cases, high percentage of identity (>90%) is only observed within the *rfb* cluster, and not in the immediate surrounding segments (Fig 4B), nor in other parts of the whole genome (Fig 4C), a clear sign that a large extension of the *rfb* cluster is being horizontally transferred.

Certain genomic rearrangements, including inversions, can also be related to HGT (Oliveira et al, 2017), further highlighting the relevance of the above-mentioned inversions observed in *L. noguchii* strains (Fig 3C). And lastly, even though the positions of the *rfb* clusters have a slight variation among Chr1 from different

*L. noguchii* strains and *Leptospira* species (Figs 3C and 4C), they do locate at restricted positions, both when they are on the sense strand at position 1, or on the antisense at position 2. Considering all the evidence together, our data strongly suggest that genes within the *rfb* cluster are horizontally transferred among different strains and species of *Leptospira*.

### *Leptospira* strains with identical/similar serologic identity display identical/similar *rfb* cluster gene composition

Considering that the *rfb* cluster uncovers clear signs of horizontal transfer, and comprises a great genetic variability, we then wished to explore whether the specific *rfb* cluster present in a given strain

serves as a genetic signature underlying serovar identity. A number of observations support such hypothesis. Four of the Uruguayan strains for which the serogroup identity could not be assigned—because of undetectable agglutination by standard serogroup-specific antisera panels (Zarantonelli et al, 2018)—presented remarkably similar *rfb* clusters, inviting to posit that they may belong to the same serogroup/serovar (Fig 3). On the contrary, strains IP1705032 and IP1709037, both belonging to serogroup Autumnalis, were indeed grouped together according to their *rfb* gene composition (Fig 3A). It must be stressed, however, that the Uruguayan strains have not been assigned to specific serovars yet (Zarantonelli et al, 2018). We thus extended this analysis using genomic data from a wide diversity of strains of different *Leptospira* species with known serovar identities, indeed confirming that the serovar/*rfb* identity link holds (Fig 5A). By constructing matrices where presence/absence of *rfb* genes are crossed with different strains, serovar-specific patterns or signatures were unambiguously uncovered (Fig 5).

Very few genes were conserved in all the *rfb* clusters from different serovars, consistent with the previous observations (Table S6). A detailed analysis by serogroup unveiled several important observations: (i) a few serovars belonging to the same serogroup showed indistinguishable patterns (e.g., Bratislava versus Lora, Ceylonica versus Javanica, Copenhageni versus Icterohaemor-rhagiae, and Lai versus Naam); extreme relatedness had been previously reported for Copenhageni/Icterohaemorrhagiae (Santos et al, 2018), where only one indel frameshift in a single LPS biosynthesis gene explains their differentiation; (ii) differential profiles were evident within most serogroups, featuring genes that may discriminate serovars; (iii) *L. noguchii* strains did not show similar patterns to other species with well-typed serovars, not even to those belonging to shared serogroups, suggesting that these *L. noguchii* strains may represent novel serovars; and (iv) the four non-agglutinating *L. noguchii* strains showing an identical *rfb* composition (except for slight differences in IP1512017) did not share many genes with other serogroups, suggesting they perhaps belong to a new serogroup altogether.

Worth highlighting is the fact that serovars belonging to the same serogroup shared all or the vast majority of genes among their *rfb* clusters. In this regard, Australis was the most variable serogroup (Fig 5A), with strains barbudensis, bajan, and 201102933 almost clonal, and IP1611024 more closely related to serovars Bratislava and Lora. Of note, the serogroup of strain 201102933 is not known, but the proximity to barbudensis and bajan, and the clustering with other strains of serogroup Australis, strongly suggests that strain 201102933 belongs to Australis, a hypothesis-driven prediction amenable for future testing.

In further support of serovar-specific *rfb* genetic signatures is the presence of 26 *rfb* genes only present in, and shared among, those Uruguayan strains, which were not amenable to serogroup assignment (genes framed within a yellow square in Fig 5A). Almost identical gene arrays were recognized among the shared signature gene sets. After a search of orthologous protein sequences (Blast best hits' accession numbers are indicated in Table S6), among the shared signature genes several were found to encode carbohydrate-active enzymes such as UDP-glucuronate 4-epimerase [EC:5.1.3.6], GDP/UDP-N,N'-diacetylbacillosamine

2-epimerase [EC:3.2.1.184], N,N'-diacetyllegionaminate synthase [EC: 2.5.1.101], and CMP-N,N'-diacetyllegionaminic acid synthase [EC: 2.7.7.82] (using PFAM and KEGG Mapper). These enzymes likely play key roles in generating the—yet to be determined—serovar-distinctive structures of LPS *O*-antigens.

The comparative analyses described above (Fig 5A) have limitations, because not all serovars are represented. Either because of a lack of information in the characterization of isolates, or because not enough finished and good-quality draft genomes are available, information loss is inevitable, eventually hampering reliable reconstructions of the relevant *rfb* genetic clusters. To further address these issues, and as a means of testing the decisive role of gene composition in serovar determination, representative genomes corresponding to identical serovar but belonging to different *Leptospira* species were analyzed in detail. Seven reliably determined serovars were found to belong to different species. In four of them, the *rfb* clusters were split into more than one contig so that for the sake of maximum reliability, only strains corresponding to three serovars were used: Hurstbridge (from *L. broomii* and *L. fainei* strains), Hardjo (from *L. borgpetersenii* and *L. interrogans*), and Valbuzzi (from *L. interrogans* and *L. kirschneri*) (Table S5, second sheet). The gene presence/absence matrix calculated for the *rfb* clusters from this subgroup (Fig 5B) readily confirms that specific groups of *rfb* genes are associated with each serovar (Table S6, second sheet), irrespective of the species. Despite the previously stated constraint in terms of genomic fragmentation, a "pan-*rfb*" was created considering the genes resulting from the analysis in Fig 5A. The comparison of this artificial *rfb* against WGSs from different serovars, including more than one strain in each case, showed the same genetic pattern for several examples (Fig 5C and Table S6, third sheet). These findings constitute a solid starting point to define a comprehensive set of serovar-specific genetic signatures, eventually revising the current protocols for serogroup and serovar assignments, which are extremely useful in clinical and epidemiologic work.

## Discussion

*L. noguchii* strains are pathogenic members of the genus *Leptospira*, of worldwide distribution, and, together with *L. interrogans*, one of the species most involved in human leptospirosis (Vincent et al, 2019). Beyond human infections, *L. noguchii* has been detected or isolated from different hosts (Silva et al, 2007; Martins et al, 2015; Barragan et al, 2016; Zarantonelli et al, 2018), exhibiting a singular host adaptation capacity, and being one of the few *Leptospira* species isolated not only from different mammals but also from amphibians (Everard et al, 1988; Gravekamp et al, 1991). However, *L. noguchii* is still a poorly characterized species, with scarce information about the circulating serovars. A few draft genomes of *L. noguchii* have been published (Moreno et al, 2015; Nieves et al, 2019), but no complete genomes had been reported so far.

The 12 *L. noguchii* strains that we have now sequenced were selected such that a broad range of serogroup variants were included, and different hosts including cattle, human, and amphibians. These genomes were analyzed in terms of their core

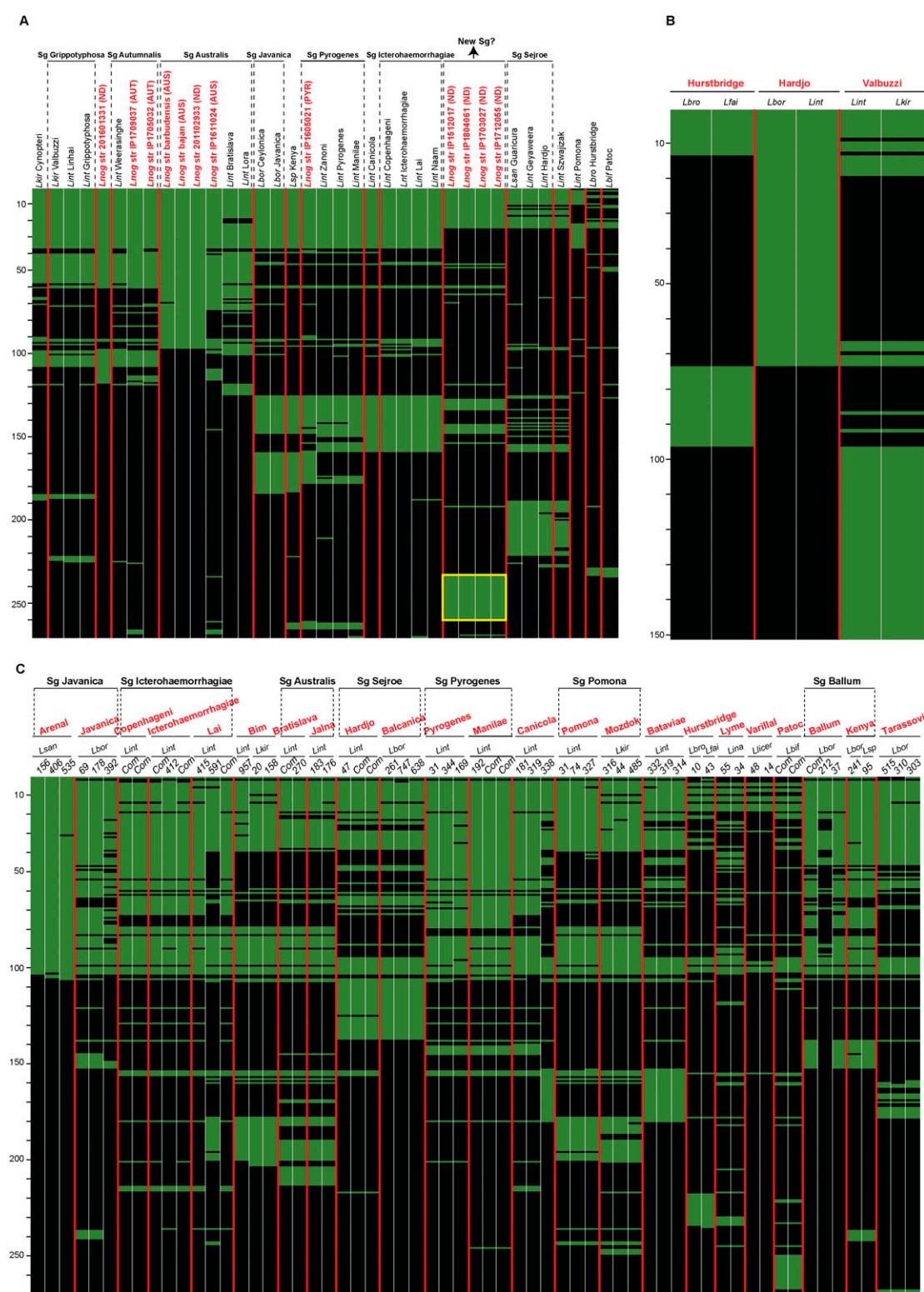

**Figure 5. Gene presence/absence matrices of *rfb* clusters from different *Leptospira* strains and species, covering a range of distinct serogroup/serovar identities.**
Horizontal lines correspond to individual genes or set of genes grouped according to their percentage of similarity cutoff (60%), green meaning presence, and black absence. Scales on the left of the matrices indicate the number of different genes being compared. Columns correspond to different *Leptospira* spp. serovars as indicated on the columns' labels. **(A)** Strains were organized after hierarchical clustering considering presence/absence of *rfb* genes. Strain names marked in red are those sequenced in this study, and their serogroup identity is indicated between parentheses. Serogroups comprising several serovars are indicated in dotted brackets and bold lettering. The yellow square indicates genes exclusively present in non-agglutinating *L. noguchii* strains. **(B)** Representation of three serovars (labeled in red), each

genome and strain-specific genes and put in a broader context by studying their phylogenetic relationship with other *L. noguchii* strains. Special attention was given to analyzing the genetic composition of the *rfb* cluster, which is notoriously linked to serovar determination (Patra et al, 2015; Picardeau, 2017). Moreover, this analysis was extended to other serovars from different *Leptospira* species.

The genome architecture in terms of replicon content and its organization is overall similar to that found in other *Leptospira* species (Picardeau et al, 2008), exhibiting two main chromosomes and a diverse repertoire of plasmids. On average, *L. noguchii* has a larger genome (4.8 Mb) compared with other relevant pathogens such as *L. interrogans* (4.6 Mb) and *L. borgpetersenii* (3.9 Mb), with a larger array of accessory genes that might be involved in *L. noguchii*'s remarkable adaptability. Indeed, ~30% of the genes from the strains sequenced in this study correspond to accessory genome, with a clear enrichment in carbohydrate biochemistry pathways. Concerning the plasmid repertoire, some strains showed identical or nearly identical protein-encoding plasmidic genes (Fig 1), suggesting that these replicons may be transferred between strains. Such exchange among different species may be evolutionarily ancient, considering the large number of shared proteins that can be identified in several cases (e.g., *L. mayottensis* str 200901116 p1, *L. mayottensis* str MDI222 pLmayMDI222, *L. interrogans* sv Bataviae str 1489 p4, and *L. noguchii* str IP1611024 p1). On the contrary, plasmid-borne gene variability seems larger in some *Leptospira* species (Table S3, first sheet), perhaps reflecting a varying contribution of horizontal plasmid acquisition as a source of adaptation. COG analyses revealed no variation in functional representation between plasmids from *L. noguchii* versus those from other species, except for the absence of genes related to amino acid metabolism and transport in *L. noguchii* plasmids.

Phylogenetic analyses using *L. noguchii* WGS data (Fig 2) did not reveal a correlation of genotype with geographical distribution nor with host specificity traits. This is consistent with previous reports (Loureiro et al, 2020), now further including the set of 10 finished genomes and two high-quality drafts that were not previously available. This led us to focus on the study of genes associated with LPS synthesis, to explore genotype–phenotype relationships that could underlie the rich phenotypic complexity exhibited among and within *Leptospira* species, which is expressed as disparate serovars/serogroups. It is known that the expression of surface epitopes in the LPS is a major determinant for serovar identity, particularly concerning the LPS's composition and spatial arrangement of sugars (Adler & de la Peña Moctezuma, 2010). The *rfb* cluster, which harbors most of the genes encoding enzymes for the LPS *O*-antigen synthesis, has been studied in *Leptospira* species of different serovars (de la Peña-Moctezuma et al, 1999; Kalambaheti et al, 1999; Bulach et al, 2000; Casas et al, 2005; Fouts et al, 2016), showing great variability in terms of genetic content and synteny. LPS *O*-antigens are frequently synthesized through a so-

called Wzx/Wzy-dependent pathway, implicating an *O*-flippase (Wzx) and an *O*-polymerase (Wzy). This biosynthesis system is complex and includes several proteins highly conserved in Gram-negative and Gram-positive bacteria that possess glycan polymers on their cell surfaces, such as LPS *O*-antigen, spore coats, enterobacterial common antigen, and outer capsules (Islam & Lam, 2014). *Leptospira* spp. appear to conserve the Wzx/Wzy-dependent *O*-antigen biosynthesis pathway, exhibiting orthologues of the genes encoding these two enzymes within the *rfb* cluster (Nascimento et al, 2004), even though variations of the typical pathway are anticipated based upon the sporadic presence of *wzy* and *wzz* genes (Fouts et al, 2016). However, a definite association between the presence of genes and serovar determination has not been shown. Some studies in *Leptospira* have been insightful, albeit focused on identifying pathogenicity determinants via characterization of LPS products (Patra et al, 2015; Vanithamani et al, 2021), but limited to the examination of subsets and not all the genes present within *rfb* clusters.

By including 12 genome sequences from *L. noguchii*, and systematically extending the analysis to all *Leptospira* species with assigned serovar identity, we have now substantiated a definite and biunivocal link between *rfb* gene composition and serogroup/serovar identity (Fig 5). The requirement of closed genomes in studies that depend on the analysis of gene presence/absence cannot be overstated, as even minor assembly mistakes, often introduced in multiple contig draft genomes, can lead to major misinterpretations. Many available draft genomes suffer from incompleteness artifacts because of contig closure errors (Kosugi et al, 2015; Lu et al, 2020). Progress in the elucidation of the structure of different LPS variants from *Leptospira* can be anticipated using NMR, as done with other bacterial genera (Fontana et al, 2014). Such an approach critically depends on accurate WGS information, to single out ambiguous alternatives from the carbohydrates' NMR spectra according to the specific set of glycosyltransferases present in the genome. The LPS structures shall thus be conclusive about their dependence on distinct sets of genes present in the genomes, rather than on regulation of gene expression. A serovar-specific genetic fingerprint such as the one we are now reporting shall be instrumental to shifting from serologic techniques to simpler and more accurate PCR-based serovar determination protocols. Considering the clinical impact that different *Leptospira* serovars exhibit, associated with distinct host adaptation and virulence phenotypes, such molecular genetic approaches have been attempted, but only at the level of serogroup (Cai et al, 2010) or for some serovars (Bezerra da Silva et al, 2011; Medeiros et al, 2022) suffering from considerable cross-detection among strains. As further and reliably complete *rfb* gene clusters from finished/closed genomes become available, more precise gene composition assessments among *Leptospira* serovars are expected to be made.

one corresponding to two different *Leptospira* species as marked, is shown side by side with the same matrix representation as in panel (A). **(C)** Comparison of the *rfb* cluster in whole genomes from different species (Lsan, *L. santarosai*; Lint, *L. interrogans*; Lkir, *L. kirschneri*; Lbor, *L. borgpetersenii*; Lbro, *L. bromii*; Lfai, *L. fainei*; Lina, *L. inadai*; Llicer, *L. liceriasae*; Lbif, *L. biflexa*; Lsp, *Leptospira* sp.), presented side by side by grouping species with the same serovar (labeled in bold red) and serogroup (enclosed in dotted brackets and bold lettering). The numbers at the top of each column correspond to the number of contigs for draft genomes, whereas complete/finished genomes are marked as *Com*.

Added to this serovar-specific signature found in the *rfb* cluster, strong indications that the *rfb* functions as a large genomic island, dispersed via events of HGT, were uncovered. Hallmarks of HGT include (i) the characteristic low GC content within the *rfb* clusters, ostensibly distinct from its flanking DNA segments, (ii) signs of genomic rearrangements (inversions) in the cluster's surroundings, and (iii) the fact that identical/nearly identical *rfb* clusters were found in different *Leptospira* species. A similar HGT scenario of genomic islands has been described in different classes of Proteobacteria, concerning LPS(*O*-antigen)- and capsule polysaccharide(K-antigen)–encoding loci (Thrane et al, 2015; Bian et al, 2020; Huszczynski et al, 2020; Buzzanca et al, 2021). Such loci have been observed to locate at highly plastic regions of the genomes from enterobacteria such as *Escherichia* and *Salmonella*, *Pseudomonas*, *Vibrio*, and *Aliarcobacter*, among other genera, exhibiting clear evidence of HGT underlying locus exchange.

Taking all the evidence together, our results strongly suggest that serovar identity can change in *Leptospira* by HGT of a part or even the entire *rfb* cluster, acting as an LPS *O*-antigen–encoding genomic island. This HGT phenomenon seems to occur within and among species from the *Leptospira* genus, contributing to population diversity and adaptability. This observation is consistent with the large variation of gene composition among different *rfb* clusters, with more or less genes being transferred in different cases, together with the ill-defined downstream limit of the island that had already been reported (Fouts et al, 2016). The molecular HGT mechanism explaining *rfb* exchange in *Leptospira* remains to be determined, potentially by homologous recombination, IS elements—which are indeed observed surrounding and within the locus—or phages (Wang & Quinn, 2010).

# Materials and Methods

### DNA extraction, sequencing, and assembly

Genomic DNA was extracted from 100 ml of a $10^8$ bacteria/ml culture of each isolate using QIAGEN Genomic-tip 100/G and Genomic DNA Buffer Set (QIAGEN). PacBio SMRT sequencing was performed with RSII technology (McGill University/Genome Quebec; Eurofins). De novo assembly was performed with HGAP v.4 (Chin et al, 2013) available on SMRT Link v.7 (default parameters, except, min. subread length: 500; estimated genome size: 4.8 Mb), Canu (https://github.com/marbl/canu) (Koren et al, 2017), Unicycler (https://github.com/rrwick/Unicycler) (Wick et al, 2017), or Trycycler (https://github.com/rrwick/Trycycler) (Wick et al, 2021). The polishing step was run on SMRT Link v.7 using the Resequencing application (default parameters).

### Genome data sets

Genomes included in the phylogenetic analyses were downloaded from GenBank (https://www.ncbi.nlm.nih.gov/genbank/) or from the Institut Pasteur Bacterial Isolate Genome Sequence Database (https://bigsdb.pasteur.fr/leptospira/). The metadata for all isolates, including for those sequenced in this study, are summarized

in Table S4. Genomes from strains of known serovar used for the *rfb* cluster comparisons were downloaded from Patric (https://www.patricbrc.org), and their metadata are summarized in Table S5. Plasmid sequences from other *Leptospira* species were also obtained from GenBank, and their associated metadata are summarized in Table S3, along with the general features of plasmids from the strains sequenced in this study. Of note, plasmids p2 from strain "barbudensis" and p5 from "IP1712055" were not included in the network analysis (Fig 1 and Table S3, third and fourth sheets) to simplify the comparison, as they show high similarities with plasmids p1 and p3 from those strains, respectively. Nevertheless, all GenBank plasmid sequence files are included as Supplemental Data 2.

### Phylogenetic analyses

All 38 genome sequences (including *L. interrogans* str. 56601 and *L. kirschneri* str. 200702274 as outgroups) were annotated using Prokka version 1.13.7 (Seemann, 2014). Orthology between the coding sequences was inferred using the combination of the two algorithms COG and OMCL through GET_HOMOLOGUES version 20190411 (Contreras-Moreira & Vinuesa, 2013). The sequences of orthologous genes that are single copy and corresponding to the softcore (sequences present in more than 95% of the genomes) were aligned using MAFFT version 7.407 (Katoh & Standley, 2013). The resulting alignments were filtered using BMGE version 1.12 (Criscuolo & Gribaldo, 2010) and concatenated in a partitioned supermatrix using AMAS (Borowiec, 2016). The best-fit model of each partition and the maximum-likelihood phylogeny was performed using IQ-TREE version 1.6.11 (Nguyen et al, 2015) and 10,000 ultrafast bootstraps (Hoang et al, 2018). The same protocol was followed for the construction of the phylogenetic tree, but sequences were codon-aligned using TranslatorX version 1.1 (Abascal et al, 2010).

The ANI index was calculated using the OrthoANIu algorithm available at EzBioCloud (https://www.ezbiocloud.net/tools/ani) (Yoon et al, 2017).

### Genomic analyses

Comparative pangenome analysis was performed using Roary version 3.11.2 (Page et al, 2015). By combining the use of Blast+ (Camacho et al, 2009) and KEGG (Aramaki et al, 2020; Kanehisa & Sato, 2020), it was possible to assign functions of selected core and accessory genes (Table S2). *Leptospira* genomes with assigned serovars and less than 500 contigs (Table S5) were downloaded, and their *rfb* clusters, lipid A, and core oligosaccharide biosynthesis–encoding clusters were analyzed. Location of sites and gene cluster sequence extraction were done with the bioinformatics tools included in Emboss 6.6.0 (Rice et al, 2000) and then annotated using Prokka version 1.13.7. Softcore *rfb* alignments were performed with MAFFT through Roary version 3.11.2, considering 60% identity and gene presence in at least 60% of strains included in the analysis. The resulting alignment was then used to calculate the phylogenetic tree using IQ-TREE version 1.6.11. The synteny of *rfb* clusters, and pairwise genome comparisons to determine conservation, was inferred and represented using Easyfig

2.2.2 (Sullivan et al, 2011). Gene presence/absence analyses among *rfb* clusters from different serovars (Supplemental Data 3) were performed by protein-level cross-matching and subsequent network associations. Briefly, a pairwise comparison of each *rfb* cluster with one another was conducted using Blastp. The network connection was thereafter established using the previously generated Blast files and NetworkX version 2.6.2 (Hagberg et al, 2008) with 60% similarity threshold; thus, proteins having ≥60% similarity were grouped together generating the gene presence/absence matrices. Close-up plots of GC content along linearized sequence fragments were performed with DNAPlotter (Carver et al, 2009). Network association analysis of plasmidic protein–encoding gene repertoire was carried out as described for the *rfb* cluster, considering a 60% similarity cutoff. Hierarchical clustering according to the shared protein–encoding genes (options used: Euclidean distance, ward linkage) was performed using available tools at https://mev.tm4.org. Functional annotation was done using eggnog-mapper v2 (Huerta-Cepas et al, 2017). Transposase positions across Chr1 of *L. noguchii* strains were obtained from the Prokka annotation and then represented using the online version of shinyCircos (https://venyao.xyz/shinycircos/) (Yu et al, 2018).

## Data Availability

The genome sequences have been deposited in DDBJ/ENA/GenBank under the BioProject PRJNA803166, specifically with the following accession numbers: *L. noguchii* strain barbudensis, CP091967–CP091970; *L. noguchii* strain 201601331, CP091962–CP091966; *L. noguchii* strain IP1512017, CP091957–CP091961; *L. noguchii* strain IP1605021, CP091953–CP091956; *L. noguchii* strain IP1611024, CP091947–CP091952; *L. noguchii* strain IP1703027, CP091943–CP091946; *L. noguchii* strain IP1705032, CP091940–CP091942; *L. noguchii* strain IP1709037, CP091936–CP091939; *L. noguchii* strain IP1712055, CP091928–CP091935; *L. noguchii* strain IP1804061, CP092112–CP092116; *L. noguchii* strain bajan, JAKNBP000000000; and *L. noguchii* strain 201102933, JAKNBO000000000.

## Supplementary Information

## Acknowledgements

This research was supported by Institut Pasteur through grant PTR 30-2017 (M Picardeau, A Buschiazzo, and FJ Veyrier); by Institut Pasteur & Institut Pasteur de Montevideo through their Pasteur International Joint Research Units program "Integrative Microbiology of Zoonotic Agents" grant IMiZA-2017 (M Picardeau and A Buschiazzo); and by the Natural Sciences and Engineering Research Council of Canada discovery grant (RGPIN-2016-04940) (FJ Veyrier). C Nieves received a Ph.D. studentship Calmette & Yersin from the Institut Pasteur International Network. FJ Veyrier received a Junior 1 and Junior 2 research scholar salary award from the Fonds de Recherche du Québec—Santé. The funders had no role in study design, data collection and analysis, decision to publish, or preparation of the article. We thank Howard Takiff and Lizeth Caraballo (Venezuelan Institute of Scientific Investigation) for providing a *L. noguchii* isolate; and Gregorio Iraola and Ignacio Ferrés for helpful discussions.

## Author Contributions

C Nieves: conceptualization, data curation, formal analysis, validation, investigation, visualization, methodology, and writing—original draft, review, and editing.
AT Vincent: data curation, formal analysis, investigation, methodology, and writing—review and editing.
L Zarantonelli: resources and writing—review and editing.
M Picardeau: resources, funding acquisition, and writing—review and editing.
FJ Veyrier: conceptualization, data curation, formal analysis, supervision, funding acquisition, methodology, and writing—review and editing.
A Buschiazzo: conceptualization, formal analysis, funding acquisition, validation, visualization, and writing—original draft, review, and editing.

## Conflict of Interest Statement

The authors declare that they have no conflict of interest.

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
