## [Reviewer comments · Life Science Alliance]

Life Science Alliance

Horizontal transfer of the rfb cluster in *Leptospira* is a genetic determinant of serovar identity

Cecilia Nieves, Antony Vincent, Leticia Zarantonelli, Mathieu Picardeau, Frederic Veyrier, and Alejandro Buschiazzo
DOI: <https://doi.org/10.26508/lsa.202201480>

Corresponding author(s): Alejandro Buschiazzo, Institut Pasteur de Montevideo and Frederic Veyrier, Centre Armand-Frappier Sante Biotechnologie, Institut National de la Recherche Scientifique, Universite du Quebec

Review Timeline:

Submission Date:	2022-04-12
Editorial Decision:	2022-06-10
Revision Received:	2022-10-01
Editorial Decision:	2022-11-13
Revision Received:	2022-11-22
Accepted:	2022-11-23

Scientific Editor: Novella Guidi

Transaction Report:

June 10, 2022

Re: Life Science Alliance manuscript #LSA-2022-01480-T

Dr. Alejandro Buschiazzo
Institut Pasteur de Montevideo
Unit of Protein Crystallography
Institut Pasteur de Montevideo
Mataojo 2020
Montevideo 11400
Uruguay

Dear Dr. Buschiazzo,

Thank you for submitting your manuscript entitled "The genetically variable rfb locus in *Leptospira* is a mobile cassette and a molecular signature of serovar identity" to Life Science Alliance. The manuscript was assessed by expert reviewers, whose comments are appended to this letter. We invite you to submit a revised manuscript addressing the Reviewer comments.

Thank you for this interesting contribution to Life Science Alliance. We are looking forward to receiving your revised manuscript.

Sincerely,

B. MANUSCRIPT ORGANIZATION AND FORMATTING:

Reviewer #1 (Comments to the Authors (Required)):

The manuscript presents the genomic comparison of WGS of *L. noguchi* strains isolated from different hosts from Uruguay. To my understanding this is the first report where a deeper analysis of *noguchi* strains is performed and for that reason the results obtained here are of major relevance. Many comparative genomic works in *Leptospira* are related to other species diminishing the importance of other agents responsible as well for Leptospirosis. Also the relevance of data regarding the differentiation of Sv is extremely useful for the leptospirosis community, but although the work is presented correctly and the methodologies and analyses realized are the proper ones, I have some issues that should be addressed before publication mainly regarding the conclusions arised and the way that the results are presented.

Major concern

The authors highlighted that *rfb* locus is a mobile cassette in *L. noguchi* broadening this conclusion to all the different species analysed and in fact this is included in the title of the manuscript, but this conclusion is raised from a very scarce evidence as is pointed out in Line 364 where they themselves said that this "cannot be ruled not even could be reliably supported". Although they showed a marked change in GC content that is ONE of the hallmarks for mobile elements, another items should be present: why did the authors do not showed the flanking regions with the IS elements that they said that were present?, and if IS elements were found are the transposases found in the *rfb* locus functional? These should be carefully considered before claim that this important locus is a mobile cassette. Also for this reviewer is not so clear that there is a defined preferential site for all the species in the genus according to what can be seen in the Sup Fig S5 where a broad location of the locus is observed. The *Leptospira interrogans* the GC content does not seem to be very different from the surrounding region as well. These conclusions are maybe right for the 7 *noguchi* strains analysed, but not for the other ones. For that reason, I consider that there is insufficient data supporting that *rfb* is a mobile cassette and further exp are needed to sustain this, ie analyse a larger number of genomes from all the species focusing in flanking regions and transposases structure, etc. Till that the title should be changed leaving only that *rfb* is a molecular signature for sv identity.

Fig 4. This is indeed a very complex figure to follow. The lines representing each gene of the *rfb* locus are very diffused and it is not clear how many genes are involved in each line, the authors said that some Uruguayan strains presents a differential set of genes that are absent in others strains making this a hallmark for the sv identity of these strains, but how many genes are those ? and more importantly which genes are those?. This is the main issue in this figure and in the manuscript as represent sv identity. The authors presented the data as presence/absence of different set of genes belonging to the *rfb* locus, which is of extreme importance having in mind the main goal of the manuscript that is the differentiation or characterization of each sv, but again which are those genes it would be very interesting to see that in the figure. I consider that should not be very difficult in terms of reconstruction of the figure to add the names or maybe a number associated to each gene pointed out in an extra column in the left of the figures.

Minor comments

Fig 1. Plasmids phylogeny. The figure is a bit confusing regarding the isolates names from where the plasmids were obtained. The name of the species (not only *biflexa* and *kobayashi*) will clarify the figure apart from the countries or the SvS, as the clusters observed are related to the species (ie *noguchi*).

'Referee Cross-Comments'

I strongly agree with reviewer 3 and all his suggestion particularly those related with the mobile characteristic of *rfb* locus. As I mentioned in my comments I consider there is scarce data to support the strong conclusion that the authors has arrived. On the other hand plasmids analysis as reviewer 3 also pointed out need further analysis in the context of other *Leptospira* species, but at least in *noguchi* evolution.

The suggestions of reviewer 2 are correct.

Reviewer #2 (Comments to the Authors (Required)):

The manuscript entitled "The genetically variable rfb locus in *Leptospira* is a mobile cassette and a molecular signature of serovar identity" by Nieves C et al." has determined whole-genome sequences of twelve strains of the pathogenic species of *Leptospira noguchii*. The genome sequences have been comprehensively analyzed and have compared the rfb locus associated with LPS synthesis for devising straightforward genetic typing of *Leptospira* serovars. A serovar-specific genetic fingerprint shall be instrumental in devising PCR-based serovar determination protocols. The methods & assays adopted in the manuscript are well-explained for further studies. The introduction is relevant and the sufficient information related to the previous findings. The discussion and results are intelligible and compelling.

Specific minor comments for better understanding and improvement of the manuscript are listed below:

#Line 91: Chromosomes I and II in the text should be consistent with Table 1.

#Line 112: Full form of the "CRISPR" is missing.

#Line 225: "Synthetized" should be corrected with synthesized.

#Line 248: Prime symbol (3') should be correctly mentioned as 3'. Require this correction throughout the MS.

#Line 354: For consistency purposes, the unit of DNA size should be written in only one format throughout the MS either Mbp (if following the base-pair format as mentioned in #Line 354) or Mb (written in #Line 398).

#Some sentences are too long (e.g., #Line 265-269). Breaking them into small sentences would make it clearer.

The supplementary tables (individual excel files) are to be named as in text files.

Some important references on the rfb locus are missing and little description about it will be helpful to the readers.

a. Ren, Shuang-Xi, et al. "Unique physiological and pathogenic features of *Leptospira interrogans* revealed by whole-genome sequencing." *Nature* 422.6934 (2003): 888-893.

b. Bulach, Dieter M., et al. "Lipopolysaccharide biosynthesis in *Leptospira*." *Journal of molecular microbiology and biotechnology* 2.4 (2000): 375-380.

Reviewer #3 (Comments to the Authors (Required)):

In their manuscript, Nieves et al. aim at further characterizing the rfb locus as a signature for serovar identity in *L.noguchii*. As a first step, the authors describe the complete sequencing and assembly of the genomes of 10 new strains of this spirochaete, 8 isolated from cattle, 1 from an amphibian and one from human, and a draft sequence from two more strains, one from amphibian (5 contigs), the other from human (6 contigs). The genomes include a large set of plasmids. Overall the data add to a previous set of 37 draft and 6 (if I count correctly!) complete *L.noguchii* genomic sequences available at NCBI. The complete set of *L.noguchii* genomes is then used to estimate the conserved and pan genomes, to analyze the relationship and the gene repertoire of the plasmids and to confirm the genetic contribution of the rbf gene cluster to the serovar phenotypes. Some features of the rbf gene clusters are taken to suggest that it may behave as a mobile genomic island, as stated in the title (the term "island" would I think be more appropriate than "cassette" in view of the large size of the cluster).

Except for one or two sentences, the manuscript reads very fluently.

However upon deeper examination, especially of the figures (see more detailed comments below), I became somewhat frustrated by the limits of the analysis and the lack of details shown in the figures. The latter, for sure, need to be revised before publication.

I would recommend to shorten the introduction, while focussing more on what is known about the three main points of the paper: number of genomes available, classification in the genus *Leptospira*, number of genome sequences already available (URL <https://www.ncbi.nlm.nih.gov/data-hub/taxonomy/171/> may be sufficient, it readily shows the disequilibrium between species), need to have more fully assembled genomes to clarify the relationship between serovar and the LPS/O-antigen synthesis gene cluster, what is already established about this relationship in various *Leptospira*, and if anything, what is known about plasmids in the genus and *L.noguchii* in particular. In that context, a section of the discussion (lines 408-425) would better fit in this introduction.

As a consequence of the comments and questions below, the title, the discussion and conclusions will have to be adapted to what the data substantiate.

More specific comments and questions.

In 2016, Fouts et al. presented a comparison of the genomic structure of the rfb region of a set of one representative of pathogenic, intermediate and saprophytic species. There is a big "gap" between comparing one strain of a set of species and (here) 10 strains of the same one. Advantage and disadvantages of the two approaches should be addressed at least briefly. Why for instance, weren't even some of the complete genomes of *L.interrogans*, close relative of *L.noguchii*, included in the present analysis of the rfb segments?

In Table 1, it would be helpful for the reader to see the names of the strains as they are in the NCBI sequence files (IP...).

Plasmids are compared within the *L.noguchii* lineage and with plasmids from other strains. This analysis remains superficial and does hardly allow for the conclusion line 155-157 : "Variable in terms of size, *L.noguchii* plasmids showed similar GC content and density of transposase genes compared to those of other species, and a slightly higher percentage of hypothetical proteins (Supplemental Table S3) likely related to a poorer characterization of this pathogenic species.". %GC, for instance, varies from 32 to 42% ! Is this really "similar"? The percentage of hypothetical genes is so high, that it is indeed hard to come up with any conclusion about conservation as, by the way, stated line 169. Additional questions could have been addressed about these and other leptospira plasmids, e.g. are some identical/similar plasmids present in strains originating from different geographical locations or hosts (there are, but it would have been nice to read it in the text)? Are all the plasmids circular? Could some of them be circular prophages? Are there conjugation genes that could point towards mobility? Concerning lines 165-169, are there other categories besides metabolic functions that come out of the COG analysis? The tree doesn't tell much about a possible common backbone between some plasmids, besides all the IS, Tn and other mobile elements they may harbor. It seems to me that a network analysis based on shared families of conserved proteins/genes would have been more informative about possible conservation, especially with this high proportion of genes of unknown function.

Concerning the figures,

- in Fig. 3, the red and blue scale goes 100 to 64%. But this is hardly discernible on the lines, especially when they are blue on a red background. More appropriate colors need to be chosen (mind color blind readers!).

I also have a problem with the genes orientation. The legend says red lines connect similar genes in the same orientation, blue lines, genes in opposite orientation. But all the genes (orange arrows) point into the same direction? Including the coordinates of the chromosomal regions shown, would have made the relative orientation in the chromosome readily visible.

- I have a serious problem with Fig. 5. Looking at part A, one really have the impression that some of the chromosomes have a large inversion somewhere between "4 and 8 hrs". There are many reasons for this to happen, whether in vivo (during real life) or in silico (during sequence assembly) if identical DNA segments (IS or prophages) in opposite orientation are present on the chromosome. In vivo, homologous recombination between the segments provoke an inversion of the sequence they flank. I tried to look if this is the case, but with the information in the paper it turned out too difficult (different strain names, no coordinates of the 'islands' etc.). Such an inversion has been seen in two very closely related *L.weillii* strains CUDO6 and CUD13 (Olo Ndela et al. 2021). If this turns out to be the case here, the argument about mobility of the rbf cluster becomes quite weak!

In the same figure, part B, again the coordinates of the rbf clusters should be shown, to illustrate their relative orientations. More details about the nucleotide sequences or genes flanking the clusters would have allowed to readily exclude my alternative explanation (although I agree that, as stated, flanking genes on one side are more conserved than on the other, but at the scale used in the figure, it is hard to precisely evaluate).

Horizontal gene transfer and mobility (HGT).

While it is obvious that HGT is involved in the evolution of the rbf segment, I'm afraid the absence of any signature expected for a mobile element (conserved ends vs non conserved flanking regions, mobility genes e.g. transposases, integrases, conjugation genes, phage-like genes...) makes me really doubt this segment is really mobile. HGT may occur in vivo even if the gene cluster isn't mobile per se, by transduction, conjugation or transformation (DNA assimilation by the bacterium). Is there any indication leptospira could be naturally transformable? Therefore my wish to see more information about the possible presence of conjugative plasmids.

Most importantly, I'd need to be convinced my above hypothesis of a chromosomal inversion can be excluded before I can accept the title of the paper!

Reviewer #1 (Comments to the Authors (Required)):

*The manuscript presents the genomic comparison of WGS of *L. noguchi* strains isolated from different hosts from Uruguay. To my understanding this is the first report where a deeper analysis of *noguchi* strains is performed and for that reason the results obtained here are of major relevance. Many comparative genomic works in *Leptospira* are related to other species diminishing the importance of other agents responsible as well for Leptospirosis. Also the relevance of data regarding the differentiation of Sv is extremely useful for the leptospirosis community, but although the work is presented correctly and the methodologies and analyses realized are the proper ones, I have some issues that should be addressed before publication mainly regarding the conclusions arised and the way that the results are presented.*

Major concern

*The authors highlighted that *rfb* locus is a mobile cassette in *L. noguchi* broadening this conclusion to all the different species analysed and in fact this is included in the title of the manuscript, but this conclusion is raised from a very scarce evidence as is pointed out in Line 364 where they themselves said that this "cannot be ruled not even could be reliably supported".*

<...>

RESPONSE: we appreciate these comments and have taken them into account to revise our ms. This reviewer's comments, as well as reviewer #3's, led us to recognize we don't have yet enough direct evidence to confirm nor to rule out mechanistic hypotheses about gene mobility, and we have made a major revision of the text throughout to make this clear. Just for the purpose of discussing the reviewers' comments in detail, let us better precise about line 364 of our original submission, as reviewer #1 brings it up. Probably stating the whole sentence is more clarifying, as it said: "A putative transposition mechanism explaining *rfb* mobility and horizontal integration can thus not be ruled out, but neither can it be reliably supported."

We wish to highlight that, as the sentence stated it, this was referred to one of the several mechanisms by which mobile genetic elements can move, namely transposition. In the context of the Discussion section, it was a phrase that completed the analysis about the presence of insertion sequence (IS), transposase-like CDSs. Several ISs were indeed found flanking the *rfb* loci in *Leptospira* spp. including in the *L. noguchii* WGSs we are focusing on in this work.

Having said that, we agree in that we need to be very cautious, since, as we said on line 364 as well, "there were also ISs within the *rfb* loci, such that no clear-cut physical association could be ascribed." All in all, transposition might be explaining the *rfb*'s horizontal transfer, but further experimental evidence is needed, beyond the scope of this first paper.

*Although they showed a marked change in GC content that is ONE of the hallmarks for mobile elements, another items should be present: why did the authors do not showed the flanking regions with the IS elements that they said that were present?, and if IS elements were found are the transposases found in the *rfb* locus functional? These should be carefully considered before claim that this important locus is a mobile cassette.*

RESPONSE: New Figure 4 and its accompanying text, now analyses in greater detail the flanking regions of the *rfb* cluster, both within *L. noguchii* (different strains/serogroups/variants), as well as comparing different *Leptospira* species. The main conclusion of this is that indeed the *rfb* cluster is highly conserved (acting as an island) in species with similar serovar identities, whereas the cluster's flanking regions are not: a further strong indication of the horizontal transfer of the *rfb* cluster of genes.

On the other hand, transposases located across chromosome 1 of the closed genomes are now shown in Supplementary Figure S4. As can be seen, the *rfb* cluster presents IS elements surrounding and within the cluster. Such elements are seen evenly distributed throughout chromosome 1. The exact coordinates of the transposases, as well as the group they belong to resulting from their annotation with Prokka, can now be found in Supplementary Table S5. We agree as mentioned in the previous point that we have not corroborated whether these transposases are indeed functional, which is needed to conclude about the cluster behaving as a mobile genetic element.

Also for this reviewer is not so clear that there is a defined preferential site for all the species in the genus according to what can be seen in the Sup Fig S5 where a broad location of the locus is observed. The Leptospira interrogans the GC content does not seem to be very different from the surrounding region as well. These conclusions are maybe right for the 7 noguchi strains analysed, but not for the other ones. For that reason, I consider that there is insufficient data supporting that rfb is a mobile cassette and further exp are needed to sustain this, ie analyse a larger number of genomes from all the species focusing in flanking regions and transposases structure, etc. Till that the title should be changed leaving only that rfb is a molecular signature for sv identity.

RESPONSE: New Supplementary Figure S5, which shows chromosome 1 from different *Leptospira* species in a linear way, clearly reveals the existence of two predominant sites, one located at ~ 1.75 Mb, and the second one at ~ 2.50 Mb. Although both positions are not exact, the quantitative variation is the range of $\leq 15\%$ (and in most cases smaller) with respect to the chromosome size, hence far from random, and with the *rfb* consistently falling within one site or the other. To give a more precise picture, and in line with reviewer #3's comments, these two sites show signs of genome inversion (new Fig 3C) relating positions 1 and 2, at least for *L. noguchii*. Taken together, these are all further hallmarks of HGT as now explained in the ms.

As for GC content, the concern from this reviewer focuses on *L. interrogans*: yet, even in this case, while to a lesser extent, the GC content deviation is still clear, and actually following an overall similar profile as the other species' *rfb* loci. This analysis has been made with greater detail in this revised version, illustrated in Fig S5.

Last but not least, besides the "restricted" location sites of *rfb* clusters in different species and serovars, and the island-profiled GC-content deviation, different *Leptospira* species harbor the same *rfb* cluster (in terms of its genetic composition) and also a single species can harbor many different types of *rfb*, correlated to their different serovar identities. Taking all the evidence together, horizontal gene transfer (HGT) seems thus to be operating for the *rfb* cluster of genes, a conclusion that the other reviewers support, reviewer #3 especially clearly.

We take the point, and the title has been modified accordingly.

Fig 4. This is indeed a very complex figure to follow. The lines representing each gene of the rfb locus are very diffused and it is not clear how many genes are involved in each line, the authors said that some Uruguayan strains presents a differential set of genes that are absent in others strains making this a hallmark for the sv identity of these strains, but how many genes are those ? and more importantly which genes are those?. This is the main issue in this figure and in the manuscript as represent sv identity. The authors presented the data as presence/absence of different set of genes belonging to the rfb locus, which is of extreme importance having in mind the main goal of the manuscript that is the differentiation or characterization of each sv, but again which are those genes it would be very interesting to see that in the figure. I consider that should not be very difficult in terms of reconstruction of the figure to add the names or maybe a number associated to each gene pointed out in an extra column in the left of the figures.

RESPONSE: we thank the reviewer for these comments. We have now added a scale next to each panel (as shown in new Fig 5), indicating the number of genes being analyzed throughout the matrix. This not only gives an idea of the number of genes being compared, but also allows the reader to better realize the “size” of each matrix line. Additionally, Table S6 now includes an extra column indicating the product name for each gene, as well as the locus tags in each genome. Genbank files of the different rfb/WGS are provided as additional material.

Concerning the question about the number of differential genes that we are analyzing among the Uruguayan *L. noguchii* strains: they are 26, as indicated in lines 307-309 in this revised version (“In further support of serovar-specific *rfb* genetic signatures, is the presence of 26 *rfb* genes only present in, and shared among those Uruguayan strains which were not amenable to serogroup assignment”. Although not all 26 of them were discussed in detail, some specific examples (between lines 310-315) are mentioned as a result of the analysis with PFAM and KEGG Mapper. To enable readers pinpointing more easily any of these selected targets, the product name of these genes was added as a separate column on Table S6, as well as the locus_tags and sequences of the *rfb*/genomes analyzed. We attempted adding gene names to the main figure, and it becomes overly crowded.

Minor comments

Fig 1. Plasmids phylogeny. The figure is a bit confusing regarding the isolates names from where the plasmids were obtained. The name of the species (not only biflexa and kobayashi) will clarify the figure apart from the countries or the SvS, as the clusters observed are related to the species (ie noguchi).

RESPONSE: Following the reviewer’s concern, we have now removed original Fig 1, and preferred performing a network association analysis (which picks up one of reviewer #3’s suggestions). The result of this analysis is now summarized in new Fig 1, where the species, strain and plasmid names are now clearly indicated. Those strains with identified serovar are also labeled.

'Referee Cross-Comments'

I strongly agree with reviewer 3 and all his suggestion particularly those related with the mobile characteristic of rfb locus. As I mentioned in my comments I consider there is scarce data to support the strong conclusion that the authors has arrived. On the other hand plasmids analysis as reviewer 3 also pointed out need further analysis in the context of other Leptospira species, but at least in noguchi evolution.

The suggestions of reviewer 2 are correct.

Reviewer #2 (Comments to the Authors (Required)):

The manuscript entitled "The genetically variable rfb locus in Leptospira is a mobile cassette and a molecular signature of serovar identity" by Nieves C et al." has determined whole-genome sequences of twelve strains of the pathogenic species of Leptospira noguchii. The genome sequences have been comprehensively analyzed and have compared the rfb locus associated with LPS synthesis for devising straightforward genetic typing of Leptospira serovars. A serovar-specific genetic fingerprint shall be instrumental in devising PCR-based serovar determination protocols. The methods & assays adopted in the manuscript are well-explained for further studies. The introduction is relevant and the sufficient information related to the previous findings. The discussion and results are intelligible and compelling. Specific minor comments for better understanding and improvement of the manuscript are listed below:

#Line 91: Chromosomes I and II in the text should be consistent with Table 1.

RESPONSE: done

#Line 112: Full form of the "CRISPR" is missing.

RESPONSE: done.

#Line 225: "Synthetized" should be corrected with synthesized.

RESPONSE: OK

#Line 248: Prime symbol (3') should be correctly mentioned as 3'. Require this correction throughout the MS.

RESPONSE: We have indeed mentioned 3' systematically. We're afraid we don't quite understand this suggestion.

#Line 354: For consistency purposes, the unit of DNA size should be written in only one format throughout the MS either Mbp (if following the base-pair format as mentioned in #Line 354) or Mb (written in #Line 398).

RESPONSE: point taken, everything has now been made consistent (adopting Mb units).

#Some sentences are too long (e.g., #Line 265-269). Breaking them into small sentences would make it clearer.

RESPONSE: Thank you for pointing this out, we have now rephrased this sentence.

The supplementary tables (individual excel files) are to be named as in text files.

RESPONSE: Supplementary Tables names have been corss-checked and made consistent throughout.

Some important references on the rfb locus are missing and little description about it will be helpful to the readers.

a. *Ren, Shuang-Xi, et al. "Unique physiological and pathogenic features of Leptospira interrogans revealed by whole-genome sequencing." Nature 422.6934 (2003): 888-893.*

b. *Bulach, Dieter M., et al. "Lipopolysaccharide biosynthesis in Leptospira." Journal of molecular microbiology and biotechnology 2.4 (2000): 375-380.*

RESPONSE: Suggestion taken; these references have now been added.

Reviewer #3 (Comments to the Authors (Required)):

In their manuscript, Nieves et al. aim at further characterizing the rfb locus as a signature for serovar identity in L.noguchii.

As a first step, the authors describe the complete sequencing and assembly of the genomes of 10 new strains of this spirochaete, 8 isolated from cattle, 1 from an amphibian and one from human, and a draft sequence from two more strains, one from amphibian (5 contigs), the other from human (6 contigs). The genomes include a large set of plasmids. Overall the data add to a previous set of 37 draft and 6 (if I count correctly!) complete L.noguchii genomic sequences available at NCBI.

The complete set of L.noguchii genomes is then used to estimate the conserved and pan genomes, to analyze the relationship and the gene repertoire of the plasmids and to confirm the genetic contribution of the rbf gene cluster to the serovar phenotypes. Some features of the rbf gene clusters are taken to suggest that it may behave as a mobile genomic island, as stated in the title (the term "island" would I think be more appropriate than "cassette" in view of the large size of the cluster).

Except for one or two sentences, the manuscript reads very fluently.

However upon deeper examination, especially of the figures (see more detailed comments below), I became somewhat frustrated by the limits of the analysis and the lack of details shown in the figures. The latter, for sure, need to be revised before publication.

I would recommend to shorten the introduction, while focussing more on what is known about the three main points of the paper: number of genomes available, classification in the genus leptospira, number of genome sequences already available (URL <https://www.ncbi.nlm.nih.gov/data-hub/taxonomy/171/> may be sufficient, it readily shows the disequilibrium between species), need to have more fully assembled genomes to clarify the relationship between serovar and the LPS/O-antigen synthesis gene cluster, what is already established about this relationship in various Leptospira, and if anything, what is

known about plasmids in the genus and L.noguchii in particular. In that context, a section of the discussion (lines 408-425) would better fit in this introduction.

As a consequence of the comments and questions below, the title, the discussion and conclusions will have to be adapted to what the data substantiate.

RESPONSE We appreciate the careful analysis, highlighting our manuscript's strengths and weaknesses.

More specific comments and questions.

In 2016, Fouts et al. presented a comparison of the genomic structure of the rfb region of a set of one representative of pathogenic, intermediate and saprophytic species. There is a big "gap" between comparing one strain of a set of species and (here) 10 strains of the same one. Advantage and disadvantages of the two approaches should be addressed at least briefly. Why for instance, weren't even some of the complete genomes of L.interrogans, close relative of L.noguchii, included in the present analysis of the rfb segments?

RESPONSE: Different species of known serovar have been included in the comparison of the *rfb* clusters, and not only *L. noguchii*. Information on the species used in these comparisons can be found in new Supplementary Table S5.

In Table 1, it would be helpful for the reader to see the names of the strains as they are in the NCBI sequence files (IP...).

RESPONSE: done. Strains are now listed as reported in NCBI.

Plasmids are compared within the L.noguchii lineage and with plasmids from other strains. This analysis remains superficial and does hardly allow for the conclusion line 155-157 : "Variable in terms of size, L.noguchii plasmids showed similar GC content and density of transposase genes compared to those of other species, and a slightly higher percentage of hypothetical proteins (Supplemental Table S3) likely related to a poorer characterization of this pathogenic species.". %GC, for instance, varies from 32 to 42% ! Is this really "similar"? The percentage of hypothetical genes is so high, that it is indeed hard to come up with any conclusion about conservation as, by the way, stated line 169. Additional questions could have been addressed about these and other leptospira plasmids, e.g. are some identical/similar plasmids present in strains originating from different geographical locations or hosts (there are, but it would have been nice to read it in the text)? Are all the plasmids circular? Could some of them be circular prophages? Are there conjugation genes that could point towards mobility? Concerning lines 165-169, are there other categories besides metabolic functions that come out of the COG analysis? The tree doesn't tell much about a possible common backbone between some plasmids, besides all the IS, Tn and other mobile elements they may harbor. It seems to me that a network analysis based on shared families of conserved proteins/genes would have been more informative about possible conservation, especially with this high proportion of genes of unknown function.

RESPONSE: We acknowledge the reviewer's comments and questions, ultimately suggesting a more thorough analysis with regards to *L. noguchii*'s plasmids. Even though this section could become the central aspect of a research paper – if we chose to focus on it–, within this particular ms, a comprehensive analysis just goes beyond the scope of our

work (or make the article exceedingly long). Having said that, we have made a major rearrangement of this section. We have included most of the reviewer's suggested topics, and responded to several questions.

Plasmids are variable in GC content, but the range of variability is comparable to that observed in other species. In any case, we do agree that it was not clearly explained. The analysis of plasmids was now approached from another angle, and this section was almost completely rewritten.

We also agree in that the huge number of hypothetical protein-coding genes in plasmids is notorious and makes comparisons difficult. The network association analysis shown in new Fig 1 gives a better idea of the genes shared between plasmids from different strains.

Following a network association analysis, new Fig 1 allows the identification of strains from different hosts and/or locations, having similar/identical plasmids. The clearest example is provided by three strains of *L. interrogans*, two of them obtained from humans (Gui44, 611), and the third from a dog (LJ178), which share two sets of identical plasmids. Regarding general features, and as can be seen in Table S3, the vast majority of the shared proteins are hypothetical, which again makes it difficult to draw conclusions. Besides, most of these plasmids are not associated with any publication, and further description of them is lacking.

Lastly, concerning the reviewer's questions about COG analysis, we have now extended this description of plasmidic genes in lines 170-177. We do acknowledge the suggestion to perform a network association analysis, which we took into account as stated in previous responses.

Concerning the figures,

- in Fig. 3, the red and blue scale goes 100 to 64%. But this is hardly discernible on the lines, especially when they are blue on a red background. More appropriate colors need to be chosen (mind color blind readers!).

RESPONSE: Thank you for this comment, apologies for not having thought of this. We have now modified color selection to orange/blue (new Fig 3).

I also have a problem with the genes orientation. The legend says red lines connect similar genes in the same orientation, blue lines, genes in opposite orientation. But all the genes (orange arrows) point into the same direction? Including the coordinates of the chromosomal regions shown, would have made the relative orientation in the chromosome readily visible.

RESPONSE: The caption has now been modified to "Homologous regions are linked with orange lines (same orientation) and blue lines (inverted regions)". Most genes have the same orientation, although a few examples (not conserved among *rfb*'s) run in the opposite direction. Most inversions fall outside the CDS, rather in the intergenic regions. The *rfb* clusters' coordinates have been added in Fig 3B.

- I have a serious problem with Fig. 5. Looking at part A, one really have the impression that some of the chromosomes have a large inversion somewhere between "4 and 8 hrs". There are many reasons for this to happen, whether in vivo (during real life) or in silico (during sequence assembly) if identical DNA segments (IS or prophages) in opposite orientation are

present on the chromosome. In vivo, homologous recombination between the segments provoke an inversion of the sequence they flank. I tried to look if this is the case, but with the information in the paper it turned out too difficult (different strain names, no coordinates of the 'islands' etc.). Such an inversion has been seen in two very closely related L. weilii strains CUDO6 and CUD13 (Olo Ndela et al. 2021). If this turns out to be the case here, the argument about mobility of the rfb cluster becomes quite weak!

RESPONSE: We greatly appreciate his observation. Chromosomes I were aligned and indeed, as the reviewer suggested, there are inversions in that region (new Fig 3C). In some cases, the *rfb* cluster is precisely located at the boundary where these inversions occur. This has all now been described with greater detail. As we have also responded to reviewer #1 above, HGT events have the potential to cause chromosomal rearrangements, which together with the other features uncovered (GC content deviation, genetic signature criss-crossing species yet instead linking serovariants, non-conservation of *rfb* flanking regions when the *rfb* clusters are extremely similar in serogroup/serovar-related pairs) constitute a cumulus of signs that indicate HGT is indeed taking place and explaining the behavior of the *rfb* cluster of genes. We agree in that we do not have enough evidence to explain the mechanism of transfer (and certainly not to postulate the the cluster is a mobile genetic element): as also responded to reviewer #1, we have accordingly modified the ms throughout.

In the same figure, part B, again the coordinates of the rfb clusters should be shown, to illustrate their relative orientations. More details about the nucleotide sequences or genes flanking the clusters would have allowed to readily exclude my alternative explanation (although I agree that, as stated, flanking genes on one side are more conserved than on the other, but at the scale used in the figure, it is hard to precisely evaluate).

RESPONSE: Following up on the previous point, indeed we believe the reviewer's suggestion is a very good one, as it helps to prove the point. Coordinates were added on Fig 3B.

Horizontal gene transfer and mobility (HGT).

While it is obvious that HGT is involved in the evolution of the rfb segment, I'm afraid the absence of any signature expected for a mobile element (conserved ends vs non conserved flanking regions, mobility genes e.g. transposases, integrases, conjugation genes, phage-like genes...) makes me really doubt this segment is really mobile. HGT may occur in vivo even if the gene cluster isn't mobile per se, by transduction, conjugation or transformation (DNA assimilation by the bacterium). Is there any indication leptospira could be naturally transformable? Therefore my wish to see more information about the possible presence of conjugative plasmids.

RESPONSE: We appreciate agreeing with the reviewer about the strong indications of horizontal gene transfer of the *rfb* cluster (also, and following up on the suggestion, we have now replaced the denomination of *rfb* 'locus' by 'cluster' of genes).

Among the characteristics mentioned by the reviewer, we consider that the conservation of flanking regions is a particularly strong one: the conservation of *rfb*-flanking regions is mostly confirmed when comparing the same species sharing serovar (Fig 3B). Yet of course, conservation within a species is larger throughout the chromosome sequence. But now we extended this analysis to more distinct and more distant species (*L. borgpetersenii* vs *L.*

interrogans, or yet *L. interrogans* vs *L. santarosai*) using strains exhibiting identical serovar/serogroup (new Figs 4B and 4C), and now it is clear that high sequence identity is observed almost exclusively within the *rfb*, strongly supporting the hypothesis of their horizontal transfer. Other elements mentioned by the reviewer were found, such as transposases around and within the *rfb* (new Fig S4), although their abundance is elevated throughout chromosome 1. We also looked for phage-like genes, but they were only detected in two of the 12 *L. noguchii* genomes.

Most importantly, I'd need to be convinced my above hypothesis of a chromosomal inversion can be excluded before I can accept the title of the paper!

RESPONSE: responded above concerning inversion of chromosomal regions, which were indeed found. As for the title we hope that it now summarizes the two important messages of our findings, agreeing in that we need to avoid a premature conclusion of the *rfb* acting as a MGE (due to the lack of a clear molecular mechanism).

November 13, 2022

RE: Life Science Alliance Manuscript #LSA-2022-01480-TR

Dr. Alejandro Buschiazzo
Institut Pasteur de Montevideo
Unit of Protein Crystallography
Institut Pasteur de Montevideo
Mataojo 2020
Montevideo 11400
Uruguay

Dear Dr. Buschiazzo,

Thank you for submitting your revised manuscript entitled "Horizontal transfer of the rfb cluster in *Leptospira* is a genetic determinant of serovar identity". We would be happy to publish your paper in Life Science Alliance pending final revisions necessary to meet our formatting guidelines.

- please address the remaining Reviewer 3's concerns
- please upload your supplementary figure as single files
- please add ORCID ID for secondary corresponding author-they should have received instructions on how to do so
- please add a summary blurb / alternate abstract to our system
- please add the Twitter handle of your host institute/organization as well as your own or/and one of the authors in our system

A. FINAL FILES:

B. MANUSCRIPT ORGANIZATION AND FORMATTING:

Sincerely,

Reviewer #1 (Comments to the Authors (Required)):

I have received the second version of the MS of Nieves et al. To my understanding the MS have been substantially improve and all my questions and suggestion have been answered, the figures have been also improve making the reading much more feasibly.

As the authors agreed to change the tittle of the MS expressing in this way the main findings of the work I understand that even though more data would be needed to further explore the genetic mobile nature of this cluster for the genus *Leptosira*, regarding the particular objectives of noguchi the MS is ready to be published

Reviewer #3 (Comments to the Authors (Required)):

The new version of the manuscript by Cecilia Nieves and coworkers, now entitled "Horizontal transfer of the rfb cluster in *Leptospira* is a genetic determinant of serovar identity" has involved a profound revision of the text, figures and Tables, taking good account of the 3 reviewer's comments. Congratulation to the authors for their careful and comprehensive reply to these numerous comments. The result is a much improved and more complete and accurate description of the data (e.g. of the plasmids, which now brings more concrete and useful information), which is close to ready for publication (I think there is some confusions between suppl. Table numbers, although it may be in the response to reviewer 1 only ? Anyway, better check Tables 5S and 6S among others).

I still have some uncertainty on the best way to formulate the conclusions about the possible mobility of the rfb gene cluster. HGT appears obvious, but I'm still puzzled about some features. If I understand Fig. 3 and 4 correctly, it appears that different rfb clusters can reside at the same (or very close) positions in different genomes. It also appears from Fig. 3B that the cluster can reside at or near the border of a chromosomal inversion. I wonder if this could be verified (manually) by precisely comparing the nucleotide sequences upstream of marR and downstream of sdcS, for instance in the 6 strains depicted on the top part of Figure 2C (by the way the figures are not numbered in the "merged" pdf file, so that I'm not so sure of this number). How to reconcile a unique location and the gene diversity of the rfb cluster ? This may be a point to be addressed in the discussion.

Last point: O'Ndela et al. observed an inversion but didn't analyze or discuss a possible mechanism responsible for its appearance...

Minor points:

Line 788-89 : %GC in whole chromosome 1 or in 100000 bp shown?

Line 791-92: idem, really 100000 bp in *L.interrogans* Geyaweera and *L.santarosai* ? It seems to show shorter region?

Line 801: What means the straight black bar in Figure 3C? No similarity within cut off?

Fig. 3A: *L.noguchi* IP1705032 AUT and *L.noguchi* IP201601331 ND: curiously the serovars of the two strains are different but the rbf clusters are very much the same? Explain...

Fig. S4 : would it be possible to group chromosomes according to the location of the rbf cluster? That would allow a more rapid comparison of the surrounding regions.

November 23, 2022

RE: Life Science Alliance Manuscript #LSA-2022-01480-TRR

Dr. Alejandro Buschiazzo
Institut Pasteur de Montevideo
Unit of Protein Crystallography
Institut Pasteur de Montevideo
Mataojo 2020
Montevideo 11400
Uruguay

Dear Dr. Buschiazzo,

Thank you for submitting your Research Article entitled "Horizontal transfer of the rfb cluster in *Leptospira* is a genetic determinant of serovar identity". It is a pleasure to let you know that your manuscript is now accepted for publication in Life Science Alliance. Congratulations on this interesting work.

DISTRIBUTION OF MATERIALS:

Again, congratulations on a very nice paper. I hope you found the review process to be constructive and are pleased with how the manuscript was handled editorially. We look forward to future exciting submissions from your lab.

Sincerely,
